# Particulate sulfur in the upper troposphere and lowermost stratosphere - sources and climate forcing

Bengt G. Martinsson[1], Johan Friberg[1], Oscar S. Sandvik[1], Markus Hermann[2], Peter F.J. van Velthoven[3] and Andreas Zahn[4]

[1]Division of Nuclear Physics, Lund University, Sweden
[2]Leibniz Institute for Tropospheric Research, Leipzig, Germany
[3]Royal Netherlands Meteorological Institute (KNMI), De Bilt, The Netherlands
[4]Institute of Meteorology and Climate Research, Institute of Technology, Karlsruhe, Germany

*Correspondence to:* B. G. Martinsson (bengt.martinsson@nuclear.lu.se)

**Abstract.** This study is based on fine mode aerosol samples collected in the upper troposphere (UT) and the lowermost stratosphere (LMS) of the northern hemisphere extratropics during monthly intercontinental flights at 8.8 – 12 km altitude of the IAGOS-CARIBIC platform in the time period 1999 – 2014. The samples were analyzed for a large number of chemical elements using the accelerator-based methods PIXE (particle-induced X-ray emission) and PESA (particle elastic scattering analysis). Here the particulate sulfur concentrations, obtained by PIXE analysis, are investigated. In addition, the satellite-borne lidar aboard CALIPSO is used to study the stratospheric aerosol load. A steep gradient in particulate sulfur concentration extends several kilometers into the LMS, as a result of increasing dilution towards the tropopause of stratospheric, particulate sulfur-rich air by tropospheric air forming the extratropical transition layer (ExTL). Observed concentrations are related to the distance to the dynamical tropopause. A linear regression methodology handled seasonal variation and impact from volcanism. This was used to convert each data point to standalone estimates of a concentration profile and column concentration of particulate sulfur in a 3 km altitude band above the tropopause. We find distinct responses to volcanic eruptions, and that this layer in the LMS has a significant contribution to the stratospheric aerosol optical depth and thus to its radiative forcing. Further, the origin of UT particulate sulfur shows a strong seasonal variation. We find that tropospheric sources dominate during the fall as a result of downward transport of the Asian tropopause aerosol layer (ATAL) formed in the Asian monsoon, whereas transport down from the Junge layer is the main source of UT particulate sulfur in the first half of the year. In this latter part of the year, the stratosphere is the clearly dominating source of particulate sulfur in the UT during times of volcanic influence as well as background conditions.

## 1 Introduction

The global mean surface temperature has increased considerably in the two last years (NOAA, Dec. 5, 2016), which followed on a 15-year period with slow temperature evolution. CMIP5 (coupled model intercomparison project) models predict stronger than observed temperature increase in this period (Fyfe et al., 2013; Fyfe et al., 2016). Reasons for these differences were sought, and the Interdecadal Pacific Oscillation connected with increased subduction and upwelling (England et al., 2014; Meehl and Teng, 2014), variations in volcanic aerosol (Solomon et al., 2011; Santer et al., 2014) and solar (Myhre et al., 2013) forcings were identified as main causes of the discrepancies. These

phenomena are all elements of natural climate variability, thus highlighting the importance of these influences for assessing the human impact on the climate (Ramanathan and Feng, 2008).

Air from the tropical troposphere containing aerosol precursor gases is lifted into the tropical stratosphere in the Brewer-Dobson circulation. Carbonyl sulfide (OCS) is the most abundant sulfur-containing gas in the atmosphere. When lofted into the stratosphere, OCS is converted to sulfur dioxide ($SO_2$) at approximately 25 km altitude aided by

UV radiation (Crutzen, 1976) and in a next step to sulfuric acid, giving rise to the background stratospheric aerosol layer. The upward tropical flow is connected with a downward flow in the extratropics, where stratospheric air flows back to the troposphere.

The stratospheric aerosol concentration varies strongly due to volcanic influence, from events like the strong 1991 Mt. Pinatubo eruption, inducing negative radiative forcing in excess of 1 W/m$^2$ (McCormick et al., 1995), to close to

background conditions at around the turn of the millennium (Bauman et al., 2003; Martinsson et al., 2005). Overall, anthropogenic influence on stratospheric aerosol is small compared to that from volcanism (Neely et al., 2013). Recent volcanic eruptions, such as in 2008 and 2009 of the Kasatochi and Sarychev in the extratropics and in 2011 of the Nabro in the tropics, significantly altered the stratospheric aerosol (Vernier et al., 2011a; Bourassa et al., 2012). These eruptions also had a profound impact on the northern hemispheric lowermost stratosphere (LMS) (Martinsson et al.,

2009) and the global aerosol optical depth (AOD) of the stratosphere (Andersson et al., 2015). After reaching the LMS, the stratospheric aerosol is transported to the upper troposphere (UT). A recent study indicates that volcanic aerosol also has an indirect climate effect by affecting the reflectance of cirrus clouds in the UT (Friberg et al., 2015).

The UT particulate sulfur, in addition, has tropospheric sources. In the tropics, convection lofts aerosol and aerosol precursor gases from low altitudes into the UT (Hess, 2005). During the monsoon season, primarily the Asian monsoon,

appreciable amounts of aerosol and gaseous aerosol precursors are lifted to the tropopause region, extending from the UT to approximately 420 K potential temperature in the stratosphere forming the Asian tropopause aerosol layer (ATAL) (Vernier et al., 2011b). The ATAL is a seasonal and regional phenomenon, which according to atmospheric modelling is strongly connected with Asian pollution sources (Neely et al., 2014). In the extratropics warm conveyor belts (WCB) (Stohl, 2001) and deep convection are the main lofting channels, the latter especially in the summer

season. Large amounts of pollutants are long-range transported across the Pacific and Atlantic oceans, where sources are affecting the background concentrations of various species over remote continents. This transport is most efficient above the boundary layer in the pressure interval 700 – 900 hPa (Luan and Jaeglé, 2013). WCBs reaching the UT have their maximum frequency in the winter (Eckhardt et al., 2004) and transport of boundary layer air into the LMS maximizes in the winter and spring (Skerlak et al., 2014), whereas deep convection dominates in the summer (Kiley

and Fuelberg, 2006).

Aerosol particles in the UT and LMS contain a large fraction of sulfur compounds, mostly sulfates (Dibb et al., 2000, Martinsson et al., 2001, Xu et a., 2001; Kojima et al., 2004). These particles also contain a considerable fraction of carbonaceous aerosol (Murphy et al., 1998; Nguyen et al., 2008; Martinsson et al., 2009), whereof a minor fraction is black carbon (Schwarz et al., 2010; Friberg et al., 2014). In some periods, particularly during spring, crustal particles

also are found in this part of the atmosphere (Papaspiropoulos et al., 2002).

In this study the concentration of particulate sulfur in the UT and LMS is investigated based on aerosol samples collected from the IAGOS-CARIBIC platform in the time period 1999 – 2014. In this period the UT and LMS aerosol was affected by volcanism from several eruptions as demonstrated in our previous studies based on this data set (Martinsson et al., 2009; Andersson et al., 2013; Friberg et al., 2014; Martinsson et al., 2014; Friberg et al., 2015; Andersson et al., 2015). The IAGOS-CARIBIC aerosol elemental concentration measurements from the LMS are taken in strong concentration gradients that are affected by mixing tropospheric air into the lowest part of the LMS. Each measurement flight results in a small number of samples, being insufficient to reconstruct the gradient. Therefore, we have frequently relied on concurrent IAGOS-CARIBIC measurements, mostly by relating the particulate sulfur measurements to ozone concentrations to express e.g. volcanic influence on the aerosol concentration (Martinsson et al., 2009). Satellite-based measurements do usually not provide specific chemical information about aerosol particles. This lack of chemical information in e.g. lidar measurements can cause a bias in LMS particle concentrations close to the extratropical tropopause from non-volcanic species such as crustal particles and enhanced signal caused by particle hygroscopic growth. Here we present standalone estimates of the stratospheric sulfur aerosol in terms of mass concentration profiles and column concentrations based on a new method, which is then used to study the AOD of the lowest part of northern hemispheric LMS and its radiative forcing. This study also comprises a discussion on the relative importance of stratospheric and the tropospheric sources to the UT of particulate sulfur, seasonal dependences, and different modes of tropospheric transport involved.

## 2 Methods

### 2.1 Sampling, analysis and classification

This study is based on measurements of particulate sulfur taken from the IAGOS-CARIBIC platform (Brenninkmeijer et al., 2007; www.caribic-atmospheric.com/), where the atmosphere is studied using modified passenger aircrafts (March 1999 – April 2002: Boeing 767-300 ER from LTU International Airways, and May 2005 – present: Airbus 340-600 from Lufthansa) during monthly sets of usually four intercontinental flights. A large number of trace gases and aerosol parameters are measured from that platform during flights in the altitude range 8.8 – 12 km, including gaseous and condensed water, $O_3$, CO, $NO/NO_y$, VOCs, greenhouse gases, halo-carbons, mercury, particle number concentrations, size distributions and elemental concentrations (Brenninkmeijer et al., 2007, Hermann et al., 2003; Schuck et al., 2009; Baker et al., 2010; Oram et al., 2012; Zahn et al., 2012; Martinsson et al., 2014; Dyroff et al., 2015; Slemr et al., 2016; Hermann et al., 2016).

Aerosol sampling from the IAGOS-CARIBIC platform in the time period 1999 – 2014 has resulted in 1198 samples analyzed for aerosol elemental concentrations. The measurements were mainly taken in the northern hemisphere (NH) extratropics and the tropics, while only a small fraction of the samples were taken in the southern hemisphere. Here the focus is on the extratropical LMS and UT of the NH. Aerosol particles with aerodynamic diameter in the range 0.08 – 2 μm were collected with a multi-channel impactor with a collection efficiency of 97% ± 4% (Nguyen et al., 2006). The typical time required to collect one sample is 100 minutes. Accelerator-based methods were used to analyze the collected samples with respect to elemental concentrations, using Particle-induced X-ray emission (PIXE) to

analyze the concentration of elements with atomic number larger than 15 (Martinsson et al., 2001). Concentrations of hydrogen, carbon, nitrogen and oxygen were investigated by particle elastic scattering analysis (PESA; Nguyen and Martinsson, 2007). Here the particulate sulfur concentrations are used. The accuracy of the analyses is estimated to 10% and the combined uncertainty in sampling and analysis is estimated to 12%. Further analytical details are found in Martinsson et al. (2014). Finally, the concentration of particulate sulfur is mostly given as a mixing ratio by normalization to STP (standard temperature (273.15 K) and pressure (101300 Pa)). When computing column concentration and AOD of the lowest 3 km of the LMS, the STP concentrations are converted to volume concentrations using the pressure and temperature of the measurement, and the altitude dependence in the LMS of the molar volume obtained from ECMWF (European centre for medium-range weather forecasts).

The dynamical tropopause (Gettelman et al, 2011) at the potential vorticity (PV) of 1.5 PVU (potential vorticity units; 1 PVU = $10^{-6}$ K m$^2$ kg$^{-1}$ s$^{-1}$) was used to classify samples with respect to tropospheric and stratospheric air. The PV along the flight track was obtained from archived analyses from ECMWF with a resolution of 1×1 degree in the horizontal and 91 vertical hybrid sigma-pressure model levels. The PV was interpolated linearly in latitude, longitude, log pressure and time to the position of the IAGOS-CARIBIC aircraft.

*2.2 Altitude*

The UT usually holds significantly lower particulate sulfur concentration than the LMS. Combined with bi-directional exchange of tropospheric and stratospheric air across the tropopause, this leads to a gradient of increasing concentration in the LMS from the tropopause. In addition, the concentration of particulate sulfur in the LMS varies due to the influence from volcanism (Martinsson et al., 2009), which has been shown to cause significant radiative forcing (Andersson et al., 2015). In order to study the particulate sulfur gradient in the LMS, fine aerosol elemental concentration measurements from the IAGOS-CARIBIC platform were used in relation to the distance between measurement position and the tropopause (Z):

$$Z = Z_a - Z_{tp} \tag{1}$$

where $Z_a$ and $Z_{tp}$ are the altitudes of the aircraft and the tropopause. $Z_{tp}$ refer to the dynamical tropopause of 1.5 PVU, which is a low limit to ensure that very little LMS air will be considered as tropospheric. The $Z_{tp}$ was obtained from the ERA-Interim data of ECMWF, whereas the altitude of the aircraft was obtained from pressure measurement which was converted to altitude using the ECMWF data. The position of the tropopause was obtained using the aircraft as the starting position. If that position is in the UT, i.e. has a potential vorticity lower than 1.5 PVU, the tropopause is found by searching upwards in the potential vorticity field. Tropopause folds in a small number of cases induce multiple tropopauses in the vertical direction, complicating the analysis of the stratospheric samples. In order to handle that problem, searches were undertaken both upwards and downwards to find the tropopause closest to the aircraft. That distance is assigned a positive value irrespective of whether the tropopause is below or above the aircraft, because positive sign indicates stratospheric air.

The study of the aerosol concentration gradient deals primarily with the LMS. The dataset, however, contains observations both in the LMS and the UT and one sample sometimes contains particles from both regions. These concentrations are connected by the exchange across the tropopause. Figure 1 shows the samples that were taken in

the UT during the entire sampling time. It is clear that the dependence on the distance from the tropopause is non-existent ($R^2 = 0.02$). Further evaluation (not shown) by normalization to seasonal average concentrations ($R^2 = 0.01$) or to individual groups of concentration data that will be explained below ($R^2 = 0.002$) did not reveal a dependence of the UT particulate sulfur concentration on the distance from the tropopause, either. It should be pointed out that the variability in distance to the tropopause is mainly caused by variability in the altitude of the tropopause, because the altitude of the measurement aircraft is fairly constant. This implies that the distance from the tropopause in Fig. 1 does not reflect the sampling altitude, and hence not an altitude-typical degree of cloud processing. The distance to the tropopause for some summer measurements was very large, due to the seasonality of the position of the tropics. Approximately one third of all stratospheric air masses transported across the extratropical tropopause reach the 500 hPa level of the atmosphere, corresponding to approximately 5000 m transport, in 4 – 5 days (Skerlak et al., 2014). This illustrates that the exchange from the stratosphere goes deep in a rather short time. Based on these observations and arguments, the UT particulate sulfur concentration is considered independent of the distance from the tropopause under presented conditions, and thus indicative of the particulate sulfur concentrations of the air mixed into the LMS.

The time required to collect one sample was subdivided into ten time intervals of equal length, where Z was computed along the flight route. For samples taken in the LMS, the sample was represented by the average distance to the tropopause of the ten time intervals. Those time intervals when the sampling was undertaken in the UT, i.e. with Z < 0, Z was set to zero, since no Z dependence of the UT particulate sulfur concentration could be identified. This implies that samples collected entirely in the UT are found at Z = 0. Some of the samples were collected both in the LMS and in the UT. In these cases, the average Z over the ten time intervals of each sample was computed with all UT parts of a sample set to Z = 0. That way all samples could be utilized to study the particulate sulfur concentration in the LMS.

### 2.3 Methodology to evaluate the S gradient around the tropopause

The particulate sulfur concentration in the LMS is investigated by linear regression in two steps. To that end we need to consider that the LMS shows a seasonal dependence induced by variability in stratospheric circulation and exchange across the tropopause over the year. Within one season differences between years can be large, mainly due to varying influence from volcanism. Further, the onset of volcanic influence on the particulate sulfur concentration can cause large variability within the season in the same year, and patchiness of young volcanic clouds can further affect the data analysis.

This study was limited to samples taken north of 30° N, where in total 765 samples are available. The highest latitude of sampling was 77° N, and 90% of the samples (5% removed each side) were taken in the range 32 - 64° N. Most of the LMS samples (95%) were taken within 3000 m from the tropopause. In terms of altitude, the results of this study therefore can be considered representative of the range 0 – 3000 m above the tropopause.

A large number of measurements is needed to obtain high statistical significance. For that, data averaged over three months were used. However, to allow for exchange of data in the analyses in order to test the stability of the results, as well as catch smooth seasonal variations in the UTLS, a seasonal overlapping technique is used. In doing so three-months-seasons shifting by one month are used, e.g. the season MAM is followed by a season AMJ where data from

April and May are used in both "seasons". Data from one season were grouped with respect to concentration levels of the different years, resulting in 4 to 5 groups of data for each season, and in total 52 groups of data from the 12 seasons to be analyzed. Some data were excluded from these analyses, which reduced the amount of data from 765 to 694. Of the excluded 71 samples, 60 pertained to periods when fresh volcanism induced strong patchiness in the particulate sulfur concentration. Eleven samples were considered outliers for other reasons, e.g. single samples affected by volcanism during a season or recent up-transport from strongly polluted regions. The remaining data of each year were tested for systematical differences. Those years where the data overlapped in particulate sulfur – height above the tropopause space, were grouped together. This way groups with varying degrees of volcanic influence were formed. Groups typical of "background" conditions were primarily based on data obtained during the 1999 – 2002 period characterized by low volcanic influence on the stratospheric aerosol (Bauman et al., 2003; Deshler et al., 2008) and data from mid-2013 to mid-2014 when the LMS is back to near-background conditions in the NH extratropics, see Table 1 for relevant volcanic eruptions. Another period that we will return to later is mid-2005 to mid-2008 when the stratosphere was affected by three tropical eruptions in 2005 – 2006: Manam, Soufriere Hills and Rabaul (Vernier et al., 2009), which also affected the NH LMS (Friberg et al., 2014). There was some variability during these years, implying that not always all of the years were grouped together. They could be grouped with other years, e.g. 2011 and 2013 often had similar concentrations and gradients during the winter and spring seasons. These groups were handled individually in the regression procedures described next, but in a context of the discussion section these groups of "moderate influence from tropical volcanism" were averaged to describe the UT aerosol along with the "background" group described above.

First linear vertical concentration gradients of particulate sulfur in the lower LMS was calculated. Ordinary linear regression (OLR) was undertaken for the 52 groups of data mentioned above of particulate sulfur concentration as a function of the altitude above the tropopause. Figure 2a shows the cumulative frequency of the coefficient of determination ($R^2$) of the 52 OLRs undertaken. $R^2$ spans 0.48 to 0.95, 72% of the groups having $R^2$ exceeding 0.6 and 50% exceeding 0.66. The deviations from the OLR models consists of scatter that does not show any trends.

When investigating the variance of the dependent variable (the particulate sulfur concentration, $C_S$) along the independent variable (altitude above the tropopause, Z) it is clear that the variance of $C_S$ increases with increasing Z. This heteroscedastic nature of the data, which is shared by most natural science data sets, could unfavorably affect in particular the offset of the OLR. In order to further investigate the effects of this problem two variable transformations of the dependent variable were tested: logarithmic and square root transformations. The logarithmic transformation turned the problem around to the other side, i.e. the logarithm of $C_S$ has large variance for small Z and small variance for large Z. The square root transformation of $C_S$, on the other hand, shows rather constant variance along the Z axis, thus making this transformation more suitable for regression. The transformation $C_S' = \sqrt{C_S}$ was applied to the data. As already pointed out, $C_S$ and Z has a linear relation implying that the following expression should be minimized with respect to slope (a) and offset (b): $(C_S' - \sqrt{aZ + b})^2$. This results in rather tedious expressions that we solved numerically for all the 52 data groups. Figures 2b and c show comparisons between the slopes and the offsets obtained by the square root transformed regression results and the OLRs. It is clear that the heteroscedastic nature of the data causes large deviations, especially in the offset.

Yet another transformation, based on forcing the regression to comply with the data for small Z, was investigated. To that end, the average concentration ($C_{S,0}$) of the data points closest to the tropopause at the average distance $Z_0$ from the tropopause was formed. $C_{S,0}$ of the 52 data groups is on average based on 13 measurements, and the minimum was 5, and the average $Z_0$ is 88 m and the largest is 273 m. The data were transformed linearly to place $C_{S,0}$ and $Z_0$ in the origin by forming $C_S' = C_S - C_{S,0}$ and $Z' = Z - Z_0$ followed by linear regression forced through the origin, i.e. $C_S' = aZ'$. Finally, the regression results are transformed back to the form $C_S = aZ + b$, where the slope a is not changed by the translation, and the offset is obtained by $b = C_{S,0} - aZ_0$. These results of the forced linear regressions are compared with the square root transformed regressions in Figures 2b and c. As can be seen, the forced linear regressions, in contrast to the OLRs, show only small deviations from the square root transformed data. Due to the more direct determination of the offset as well as the simplicity of forced linear regression compared with square root transformation, the forced linear regression method is chosen for the analyses. Thus, for each season (s) and year (y) in total 52 forced linear regressions were undertaken:

$$C_S(y, s, Z) = a(y, s)Z + b(y, s) \tag{2}$$

where a and b varies with season and the strength of the volcanic influence.

The S concentration at the tropopause and in the UT, expressed by b in eq. 2 is dependent on the stratospheric concentration which is affected by volcanism (Friberg et al., 2015). This means that the offset of the regressions is affected. This is expressed here by:

$$b(y, s) = C_{S,UT}(y, s) = a(y, s)k(s) + C_{S,UTtrop}(s) \tag{3}$$

where the first term expresses the contribution from stratospheric sources and the second that of tropospheric sources. This results in the combined equation:

$$C_S(y, s, Z) = a(y, s)Z + a(y, s)k(s) + C_{S,UTtrop}(s) \tag{4}$$

where k(s) reflects stratospheric influence on the UT particulate sulfur concentration and $C_{S,UTtrop}(s)$ is the particulate sulfur concentration of tropospheric origin.

In order to obtain estimates of k and $C_{S,UTtrop}$ a second regression for each season is undertaken where the offsets (b) are related to the slopes (a), see Fig. 3. The relative variance of the slopes was much smaller than that of the offsets (average variance ratio of 0.19). For simplicity, the variance of the slope was therefore neglected in these regressions. The groups of data of a season differ in a and b mainly due to volcanic influence. A zero slope (a = 0) would be obtained, should the stratospheric concentration become as low as the UT concentration. The UT aerosol of tropospheric origin ($C_{S,UTtrop}$) thus can be estimated as the offset of the b – a regression. The slope of that regression shows how the offset b changes with increased slope a, hence expressing the sensitivity (k) of the UT concentration to changes in stratospheric concentration. With access to these two parameters, $C_{S,UTtrop}(s)$ and k(s), the LMS concentration gradient and tropopause concentration can be estimated based on a single measurement of the particulate sulfur concentration.

The uncertainties of the forced regression results of a given season vary among the data groups. In order to account for this variability, weights are used in the regression between the offsets and slopes of a season. The weights are based on 70% double-sided student t distribution (t70%) estimates because some of the estimated $C_{S,0}$ and $Z_0$ rely on few

observations. The t70% estimate of b ($b = C_{S,0} - aZ_0$, see above) is obtained by combining the upper t70% limit of $C_{S,0}$ with the weakest t70% slope (a) and the strongest t70% slope at the lower limit multiplied with $Z_0$. The inverse of the squared t70% estimates of b obtained in this way are then used as weights in the second step regressions.

### 2.4 Lidar data from the CALIPSO satellite

The evaluation of the particulate sulfur concentrations from the IAGOS-CARIBIC aircraft was aided by the use of lidar data from the CALIOP sensor aboard the CALIPSO (Cloud-aerosol lidar and infrared pathfinder satellite observation) satellite from NASA (Winker et al., 2010), performing 15 orbits per day covering the globe from 82° S to 82° N with a repeat cycle of 16 days during nights and days. The data evaluation was based on the methodology developed by Vernier et al., (2009). Here only the night-time data of the 532 nm wavelength lidar signals were used.

The level 1 data of version 4-10 (averaged in a grid of 180 m vertically and 1x1 horizontally) were used to obtain scattering ratios, i.e. the ratio of the measured, combined air and aerosol scattering to the modeled air scattering based on molecule and ozone number concentrations from the GMAO (global modelling and assimilation office). Cloud pixels were removed by rejecting pixels of a depolarization ratio greater than 5%. This cloud mask was extended 360 m upwards to remove faint cloud residues, and pixels from underneath clouds were removed to avoid bias from cloud

absorption, see Andersson et al., (2015) for further details. Removal of cloud pixels as well as periods of instrument failure, sometimes resulted in few observations in these pixels. In order to avoid statistical noise at least 20% of the maximum possible observation data was required for any given pixel (latitude, altitude). The pixels were generated by averaging in the longitude interval 60° E to 120° E, the main longitude region of the ATAL (Vernier et al., 2015).

## 3. Results

The linear regression methodology described in the previous section was applied to particulate sulfur concentrations ($C_S$) as a function of distance from the tropopause (Z) for seasons comprising three months. Data from different years of one season were grouped according to their concentrations, resulting in four or five groups differing with respect to the $C_S$ – Z relation. The resulting groups of data from each season were modeled by forced linear regression. These model results are used in a second regression step to model seasonal influences from transport, which in turn are used

to obtain the response of the LMS and UT particulate sulfur concentrations to changes induced mainly by volcanic eruptions.

This methodology was applied to all seasons, having a duration of three months. The final product of the regression methodology, i.e. the second regression step, of all twelve overlapping seasons is shown in Fig. 3. It is clear that slopes and offsets of the season groups obtained in the first step of forced linear regression, differing in particulate sulfur

concentration related to distance from the tropopause, can readily be described by linear regressions for all twelve three-month seasons. The relation between slopes and offsets has a very strong seasonal dependence. In some seasons small changes in the LMS sulfur concentration slope is connected with a strong change in tropopause sulfur concentration. This is most pronounced for the seasons centered in February, March and April. At the other end we

find the seasons centered in September, October and November, where a change in the slope of the first regression step due to varying influence from volcanism has little effect on the tropopause concentration. To further illustrate the connection to the measurements, Fig. 4 shows the four data groups of the seasons most (Fig. 3: FMA) and least (Fig. 3: OND) susceptible to change in the tropopause concentration due to changed stratospheric concentration. In Fig. 4a the data group of least volcanic influence includes years 1999 – 2002 and 2014, and the most influenced years are 2009 and 2012 which are the springs after the Kasatochi (August 7, 2008) and Nabro (June 12, 2011) eruptions (we have no late winter/spring data after the most powerful eruption of the period studied (Sarychev, June 12, 2009) due to maintenance of the measurement aircraft). The season centered in November have the weakest volcanic influence years 1999 – 2001, 2012 and 2013 (Fig. 4b), whereas the group with strongest slope includes the three strongest eruptions of the period studied (Kasatochi, Sarychev and Nabro) a few months after respective eruption. Despite the very strong volcanic influence of the latter group, the tropopause concentration (Z = 0) remains close to that of the closely distributed tropopause concentrations of the other groups of that season.

The sensitivity of the tropopause and UT concentration to changes in LMS concentration slope is denoted k in eq. 3 and 4, which thus is obtained for each regression depicted in Fig. 3. These results are collected in Fig. 5a. The salient features of the seasonal dependence can be described by two Gaussian distributions (Fig. 5a). The maximum sensitivity appears in the late winter and early spring when the seasonal variation in tropopause altitude has its maximum in rate of upward motion (Appenzeller et al., 1996). The maximum in rate of downward motion of the tropopause appears in the fall. This delays transport from the stratosphere to the troposphere, which is reflected by a low sensitivity k. This sensitivity can, in addition, be affected by the residence time of particulate sulfur in the UT. Interestingly, the downward transport in association with the Brewer-Dobson circulation from deeper, aerosol-rich stratospheric layers through the LMS takes place in the same season as the maximum in k, thus further enhancing the stratospheric influence on the UT by the term $a(y,s) \times k(s)$ in eq. 4.

The offsets of the regression lines shown in Fig. 3 (when a = 0) expresses the case when the stratospheric and tropospheric concentrations are equal, implying that this offset expresses the particulate sulfur concentration of tropospheric origin ($C_{S,UTtrop}$) that is mixed into the LMS, see Fig. 5b. Two Gaussian distributions were used as fits to the seasonal variation of $C_{S,UTtrop}$. As for k, the seasonal variation of $C_{S,UTtrop}$ is strong. The seasonal variation of $C_{S,UTtrop}$ will be elaborated in the discussion section.

After obtaining $C_{S,UTtrop}$ and k, the data that are needed for conversion of every measurement of the concentration to an estimate of the slope and offset of the LMS concentration are available. Thus, for each individual measurement (i) in the LMS, consisting of the particulate sulfur concentration ($C_{S,i}$) and the altitude $Z_i$ above the tropopause, the slope and offset of equations 3 and 4 are obtained from:

$$a_i = \frac{C_{S,i} - C_{S,UTtrop}(s)}{Z_i + k(s)} \tag{5}$$

$$b_i = a_i k(s) + C_{S,UTtrop}(s) \tag{6}$$

The estimated slopes and offsets are shown in Figs. 6a and b, where the dots are individual measurements and the histogram monthly averages. Here measurements taken at an altitude of less than 50 m above the tropopause are not shown, because they were judged to have too small stratospheric character for an estimate of the concentration slope

in the LMS. Both the slope and the tropopause concentration are affected by volcanism (Table 1), but the relative response of the slope is much stronger than that of the tropopause concentration, see e.g. the falls of 2008 and 2009 affected by the Kasatochi and Sarychev eruptions.

Observations at various altitudes above the tropopause are difficult to compare, due to the concentration gradient in the LMS. With the estimates of the tropopause concentration and the slope in the particulate sulfur LMS concentration,
each measurement becomes an estimate of the total amount of particulate sulfur in the altitude interval investigated by integration of eq. 4. However, first the STP concentrations ($C_{S,STP} = C_S = aZ + b$) need to be converted to volume concentrations ($C_{S,V}$), which are related by $C_{S,V} = C_{S,STP} Q_{STP}/Q_V(Z)$, where Q are molar volumes. For that purpose, the altitude dependence of the molar volume from the tropopause up to 5 km above the tropopause was extracted from temperatures and pressures obtained from ECMWF for each sample. The molar volume can be expressed as $Q_V(Z) =$
$Q_V(0)e^{wZ}$, where the tropopause is at Z = 0 and w = 0.0001535 m$^{-1}$ obtained as the average of all samples. For a measurement the molar volume is $Q_m$ obtained at distance $Z_m$ from the tropopause, and the tropopause molar volume is computed by $Q_V(0) = Q_m e^{-wZ_m}$. Finally, the column concentration of particulate sulfur for the first 3 km above the dynamical tropopause of 1.5 PVU is obtained by integration of the volume concentration:

$$C_{S,col} = \int_0^{3000} \frac{Q_{STP}}{Q_V(Z)} C_{S,STP}(Z)dZ = \int_0^{3000} \frac{Q_{STP}}{Q_V(0)} e^{-wZ}(aZ + b)dZ \qquad (7)$$


The particulate sulfur column is calculated for altitudes above the tropopause (Z) in the range 0 to 3000 m, the upper limit because too few measurements (5%) were taken above that level. After integrating the column, it is also interesting to estimate the total amount of sulfur-connected aerosol. To that end, it was assumed that the aerosol consists of 75% sulfuric acid and 25% water, which is a commonly used stratospheric composition (Rosen, 1971;
Arnold et al., 1998). This means that the total sulfur column was multiplied by a stoichiometric factor of h = 4.084 to obtain the column of the sulfuric acid – water aerosol:

$$C_{A,col} = hC_{S,col} \qquad (8)$$

The measurements were taken in the northern hemispheric latitudes higher than 30° with the highest latitude of 77°, with 90% of the data in the latitude range 32 - 64° N and 70% of the data between 37 and 57° N. The northern mid-
latitudes sulfur aerosol columns are shown in Fig. 6c as monthly averages with standard errors (the few months with only one measurement available are shown without an error bar). The sulfur aerosol column of the LMS shows large variability primarily caused by volcanism. The lowest columns are found in the period 1999 – 2002, when the volcanic influence on the stratospheric aerosol was small, see Table 1 for relevant volcanic eruptions. The time period mid-2005 to mid-2008 was affected by tropical volcanism (Vernier et al., 2011a), which also caused elevated concentrations
in the NH LMS (Friberg et al., 2014). The eruptions of the extratropical volcano Kasatochi in August 2008 placed two volcanic clouds in the stratosphere (Andersson et al., 2015), one in the LMS causing strongly elevated aerosol column of the LMS that ceased by November the same year, and the other above the LMS. The latter cloud was transported downward, causing a rise of the lower LMS aerosol column in December 2008. After some influence from several eruptions of the extratropical volcano Redoubt in the spring of 2009, the eruption of Sarychev strongly affected the
northern hemispheric stratosphere from June of 2009. The eruption of the Icelandic volcano Grimsvötn in May 2011

had a strong and short impact on the northern LMS, which is reflected by a peak in June to July 2011 (Fig. 6c), before the tropical volcano Nabro reached the northern LMS in the early fall of the same year. After that eruption a gradual decrease of the aerosol load can be seen. The concentrations after mid-2013 approaches those of the period 1999 – 2002, which was close to stratospheric background conditions.

The LMS aerosol contains a significant fraction of carbonaceous material (Martinsson et al., 2009), mainly organic in nature (Friberg et al., 2014), which will add to the aerosol columns of Fig. 6c and affect the refractive index of the particles. However, this work is dealing with the sulfurous fraction, the main fraction of the stratospheric aerosol. To put the results presented in Fig. 6c into perspective, the AOD is estimated using a simplified aerosol (which thus likely is an underestimation of the AOD). Thus, the "standard" stratospheric 75% sulfuric acid – 25% water composition, the
particle density 1.669 g/cm$^3$ and the refractive index 1.44 will be used. Furthermore, particle size distribution measurements from IAGOS-CARIBIC have been taken since 2010 (Hermann et al., 2016). The changes of the size distribution induced by the moderate 2011 eruptions of Grimsvötn and Nabro were small (Martinsson et al., 2014), and agree well with previous measurements (Andersson et al., 2015) of the stratospheric background particle size distribution by Jäger and Deshler (2002). Thus, for the estimation of the AOD the latter particle size distribution was
used for the entire time period studied. For fixed composition and particle size distribution the AOD is obtained as a fixed relation to aerosol column: $AOD = fC_{A,col}$, where f contains the relations between mass and area/extinction, in this case $f = 3.29 \cdot 10^{-6}$ $m^2/\mu g$. Finally, converting the AOD to radiative forcing (RF) using the global average relation (Hansen et al., 2005; Solomon et al., 2011) of:

$$RF = -25 \times AOD; \quad in\ Wm^{-2} \tag{9}$$

to obtain an estimate of the climate influence of the sulfate aerosol of the lower LMS. The peak AOD of the Kasatochi and Sarychev eruptions is approximately 0.006, corresponding to -0.15 W/m$^2$ in regional radiative forcing of the lowest 3000 meters of the northern hemisphere LMS. Although no detailed comparisons will be made here, we find that the AOD and radiative forcing obtained from the particulate sulfur measurements show similar tendencies as satellite-based measurements (Andersson et al., 2015). The findings presented here also corroborates the findings of Andersson
et al. (2015) on the importance of the LMS for the total stratospheric AOD and radiative impact.

**4 Discussion**

The results presented here are based on measurements in the extratropical UT and the LMS of the NH, where the latter includes the extratropical transition layer (ExTL). Bi-directional exchange across the tropopause affects strong gradients in the ExTL for species having clearly different stratospheric and tropospheric concentrations (Hoor et al.,
2002), such as particulate sulfur (Martinsson et al., 2005). In the previous section, the UT concentration, the gradient in the LMS and column amount of particulate sulfur in the ExTL were investigated, with its seasonal dependence (Fig. 5) and the influence from volcanism (Fig. 6). In the processing of the data to obtain these results, in particular one feature stands out: the seasonal dependence of the particulate sulfur concentration from tropospheric sources that is mixed into the ExTL (Fig. 5b). It shows a broad maximum from August to December, peaking in September to
November, and a deep minimum in the late winter and early spring (February and March).

Let us now compare these concentrations in the UT of tropospheric origin ($C_{S,UTtrop}$) with average concentrations of particulate sulfur in the UT for two cases: "background conditions" dominated by data from mid-1999 to mid-2002 and "moderate influence from tropical volcanism" dominated by data from mid-2005 to mid-2008, see section 2.3 for details. The seasonal dependence of these two categories are shown in Fig. 7a together with $C_{S,UTtrop}$. It is clear that, in line with the findings of Friberg et al. (2015), the UT particulate sulfur concentration is affected by volcanism. In Fig. 7a we see that the main differences in UT concentrations of the two cases appears from January to July, coinciding with the season of transport down from the Junge layer into to the LMS and the shrinkage of the LMS due to tropopause upward motion (Appenzeller et al., 1996; Gettelman et al., 2011). In the period September to November the influence from volcanism on the UT particulate sulfur concentrations is small, compare the background and moderate volcanism cases (Fig. 7a). Comparing these cases to the concentration of particulate sulfur found to be of tropospheric origin it is clear that there is a strong agreement between all three categories in the fall months, whereas differences are large during the remainder of the year. The UT particulate sulfur concentration of stratospheric origin can be estimated by subtracting $C_{S,UTtrop}$ from the two cases of UT concentrations. The results are shown in Fig. 7b, where peak stratospheric influences of 19 and 36 ng/m³ STP are found in the spring, and minimum contributions of approximately 1 ng/m³ STP in the fall for the "background" and "moderate volcanism" cases. Summing the observations up (Fig. 7c), a clear seasonal dependence in the fraction of the UT particulate sulfur concentration originating in the stratosphere was found, from close to 100% in late winter/spring to approximately 10% in the fall. On a yearly average the fraction of the UT particulate sulfur that originates in the stratosphere is approximately 50% during background conditions and 70% during moderate influence from tropical volcanism.

The tropospheric source of UT particulate sulfur could be transported from the planetary boundary layer either in the form of particulate sulfur or precursor gases, in the latter case primarily sulfur dioxide ($SO_2$). In fall, winter and spring, with maximum in the winter, vertical transport by WCBs from the boundary layer to the UT is strong (Eckhardt et al., 2004), whereas in the summer deep convection is the most important mode (Hess, 2005; Kiley and Fuelberg, 2006) in the extratropics. For sulfate, the most common chemical form of particulate sulfur, the concentration usually shows a rapid decline with altitude in the troposphere (Heald et al., 2011) associated with formation of precipitation. Cloud processing also tends to strongly reduce $SO_2$ concentrations with altitude, where the relative availability of $SO_2$ and hydrogen peroxide ($H_2O_2$) is important for the $SO_2$ lifetime in the cloud.

We have found some very distinctive characteristics of the particulate sulfur concentration of tropospheric origin in the UT, with low concentrations in February and March, increasing concentrations during the summer and maximum in the sources of tropospheric origin in the fall. Carbon monoxide (CO) is often used as a tracer of air pollution. Zbinden et al. (2013) found winter/spring maximum in the CO concentration in the UT of the NH in contrast to the tropospheric component of the UT particulate sulfur. This contrast can, at least in part, be explained by the oxidizing capacity in the UT. The summer abundance of the hydroxyl (OH) radical (Bahm and Khalil, 2004) induces a decline in CO (Bergamaschi et al., 2000; Osman et al., 2016). High abundance of this radical, on the other hand, and availability of $SO_2$ can lead to production of particulate sulfur. $SO_2$ measurements by the satellite-based instrument MIPAS have recently become available (Höpfner et al., 2015). These data have large uncertainties (Höpfner et al., 2015), and likely overestimate the $SO_2$ concentration (Rollins et al., 2017). However, here we only qualitatively use the seasonal

variation. The MIPAS results indicate a UT seasonal variation in the NH midlatitudes with low concentrations in December to March and the highest concentrations in June to September (Höpfner et al., 2015). Deep convection provides a rapid route upwards in the atmosphere, favoring transport of short-lived species like $SO_2$ (TF-HTAP-2010; Dickerson et al., 2007). Low abundance of both $SO_2$ and OH in the NH UT thus could explain the weak tropospheric contribution to the UT particulate sulfur during late winter and early spring. The increase in $C_{S,UTtrop}$ during the spring and summer months coincides with increases in convective activity as well as in the concentration of both $SO_2$ and OH, thus offering a plausible explanation. However, the quantitative understanding of the SO2 transport paths and UT seasonality requires further study. Still, we need to consider the late maximum in the fall of the tropospheric source of UT particulate sulfur.

The Asian monsoon is an important feature of the global circulation in June to September, which has been found to reach deep into the lower stratosphere. This intrusion of tropospheric air into the tropical transition layer (Sunilkumar et al., 2017) has been manifested by gas phase components including water and ozone (Gettelman et al., 2004; Randel and Park 2006) and hydrogen cyanide (Randel et al., 2010). Later an aerosol layer extending from the UT to the potential temperature of 420 K in the lower stratosphere was found at 14 to 18 km altitude (Vernier et al., 2011b; Thomason and Vernier, 2013; Vernier et al., 2015). Figs. 8a-e show monthly means of the scattering ratio obtained from the CALIPSO sensor CALIOP for the months June to October 2013, when the volcanic influence was low. Formation of the ATAL can be identified in July (Fig. 8b) with maximum intensity in August (Fig. 8c). In September still a weakened ATAL can be identified (Fig. 8d), whereas in June and October the scattering in the ATAL region at 14 – 18 km altitude is very weak (Figs. 8a and e).

The altitude range of the ATAL is above the measurement altitudes of IAGOS-CARIBIC (9 – 12 km). However, the poleward circulation along isentropes bending downwards, further amplified by an extratropical cross-isentrope, downward component, brings the ATAL down to lower altitudes. The ATAL is too weak to be traced by CALIOP in the downward transport (Figs. 8 a-e). We therefore illustrate the subsidence using the eruption of the tropical volcano Nabro in the summer 2011 (Table 1), with effluents that occupied approximately the same region as the ATAL, see Fig. 3 in Bourassa et al. (2012). The effluents of this eruption were rapidly transported to the north in the TTL. Figs 8f-j actually includes two volcanic eruptions, besides the tropical volcano Nabro, the Icelandic volcano Grimsvötn injected a volcanic cloud to the tropopause region at midlatitudes which is visible in June and July 2011. The time series of the Nabro volcanic cloud in Figs. 8f-j unambiguously demonstrates the transport down to the IAGOS-CARIBIC flight altitudes in the season of the ATAL. In Fig. 6c we see a clear increase in the particulate sulfur column in September 2011, following the decline of the June-July 2011 short peak from the of Grimsvötn eruption in May 2011, thus demonstrating agreement between CALIOP and IAGOS-CARIBIC measurements of particulate sulfur. The volcanic cloud from Nabro initially resided at somewhat higher altitude than the ATAL, implying that the ATAL can be expected to reach IAGOS-CARIBIC flight altitudes somewhat earlier in the year. The transport in this season contains little particulate sulfur during periods without fresh volcanic aerosol, compare in Fig. 7 "tropical volcanism" (this category of Fig. 7 does not include the Nabro eruption, see section 2.3) with "background" and the particulate sulfur of tropospheric origin.

The downward transport allows delayed detection of the ATAL in the fall from altitudes above the measurement altitude range of IAGOS-CARIBIC in the same way as the spring detection of aerosol transported from the Junge layer, with or without volcanic influence, as shown in Fig. 7. A difference between these two seasons is that in the spring the strong source of particulate sulfur is stratospheric, whereas in the fall the source is tropospheric extending from the UT into the stratosphere. We therefore conclude that the UT particulate sulfur concentration of tropospheric origin that starts to increase rapidly in August, peaking in September to November, and is back at low concentration in January (Fig. 5b) most likely is caused by the ATAL formation from the Asian monsoon.

Model studies find similar UT – stratosphere distribution of the ATAL as the experimental studies, as well as that Asian pollution sources strongly contribute to the ATAL, and that sulfate is an important component of that aerosol layer (Neely et al., 2014). Besides this component, primary and secondary organic aerosol constituents are predicted to be important components of the ATAL (Yu et al., 2015). The present study is, as far as we know, the first observation of a chemical component of the ATAL. In agreement with modeling results we find that particulate sulfur is a component of the Asian tropopause aerosol layer. This component affects the tropopause region from July to September at 14 to 18 km altitude (Fig. 8) and the extratropical tropopause region from August to December (Fig. 5b).

**5 Conclusions**

Particulate sulfur (usually sulfate) in the upper troposphere (UT) and the lowermost stratosphere (LMS) obtained from the IAGOS-CARIBIC platform was investigated at northern midlatitudes in the time period 1999 – 2014, which covers several tropical and extratropical volcanic eruptions. The study is based on the use of linear regression models, where individual measurements in the strong gradient of the extratropical transition layer (ExTL) can be converted to an estimate of the column of particulate sulfur in a 3000 m layer above the dynamical tropopause (here defined at 1.5 potential vorticity units). The obtained time series in particulate sulfur column concentration shows distinct response to extratropical volcanism and delayed elevation of the column concentration following tropical eruptions. Assuming the stratospheric background particle size distribution and composition (75% sulfuric acid and 25% water) the AOD and radiative forcing were estimated, e.g. the peak value following the 2009 Sarychev eruptions were estimated to 0.006 and -0.15 W/m$^2$, respectively. These estimates refer mainly to the ExTL, i.e. lowest part of the LMS, thus highlighting the importance of the lowest part of the stratosphere for the overall climate impact of volcanism.

As part of this investigation the sources of UT particulate sulfur were explored. A distinct pattern emerges where tropospheric sulfur sources dominate the supply of particulate sulfur to the UT in the fall, whereas stratospheric sources dominate in January to July, the main season of transport from the Junge layer into the LMS. The smallest contributions from the troposphere are found in February and March in conjunction with low sulfur dioxide (SO$_2$) and oxidant concentrations. As the concentrations of these species increase, the UT particulate sulfur concentration of tropospheric origin increases somewhat in April to July. The particulate sulfur concentration shows a threefold increase during the fall, with maximum concentration in September to November. Making use of lidar data from the CALIPSO satellite together with the in situ measurements we find that the Asian tropopause aerosol layer (ATAL) resulting from the Asian monsoon is the cause of the increase. The ATAL is formed at 14 – 18 km altitude and extends from the UT to

approximately 420 K potential temperature in the stratosphere, with main extension in July to September. The ATAL is transported downwards, and affects the extratropical tropopause region in August to December. As far as we know, this is the first measurement of a chemical species in particles connected with the ATAL. The stratospheric and tropospheric contributions to the UT particulate sulfur concentrations thus have strong and opposite seasonal dependences. On annual average the stratospheric contribution to the UT particulate sulfur is estimated to 50% during stratospheric background conditions. During influence from moderate tropical volcanism the stratospheric fraction rises to 70%. The particulate sulfur concentration in the UT is thus to a large degree governed by the stratosphere and volcanism.

## Data availability

Data are available upon request to the corresponding author.

## Competing interests

The authors declare that they have no conflict of interest.

## Acknowledgements

We acknowledge all members of the IAGOS-CARIBIC project, Lufthansa and Lufthansa Technik for enabling the IAGOS-CARIBIC observatory. Financial support from the Swedish National Space Board (contract 130/15) and the Swedish Research Council for Environment, Agricultural Sciences and Spatial Planning (contract 942-2015-995) is gratefully acknowledged. Moreover, the German Federal Ministry of Education and Research (BMBF) is acknowledged for financing the instruments operation as part of the joint project IAGOS-D. Aerosol measurements from CALIPSO were produced by NASA Langley Research Center.

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

**Tables**

**Table 1.** Most significant volcanic eruptions for the aerosol concentration in the northern hemisphere LMS in the time period studied.

| Volcano | Date | Lat./Long. | SO$_2$ (Tg) |
|---|---|---|---|
| Manam | 27 Jan 2005 | 4° S / 145° E | 0.1[a] |
| Soufriere Hills | 20 May 2006 | 17° N / 62° W | 0.2[b] |
| Rabaul | 7 Oct 2006 | 4° S / 152° E | 0.2[a] |
| Jebel at Tair | 30 Sep 2007 | 16° N / 42° E | 0.1[c] |
| Okmok | 12 Jul 2008 | 53° N / 168° W | 0.1[c] |
| Kasatochi | 7 Aug 2008 | 52° N / 176° W | 1.7[c] |
| Redoubt | 23 Mar 2009 | 60° N / 153° W | 0.1[d] |
| Sarychev | 12 Jun 2009 | 48° N / 153° E | 1.2[e] |
| Grimsvötn | 21 May 2011 | 64° N / 17° W | 0.4[f] |
| Nabro | 12 Jun 2011 | 13° N / 42° E | 1.5[f] |

[a]Prata and Bernardo, 2007

[b]Carn and Prata, 2010

[c]Thomas et al., 2011

[d]Brühl et al., 2015

[e]Haywood et al., 2010

[f]Clarisse et al., 2012

 **Figures**

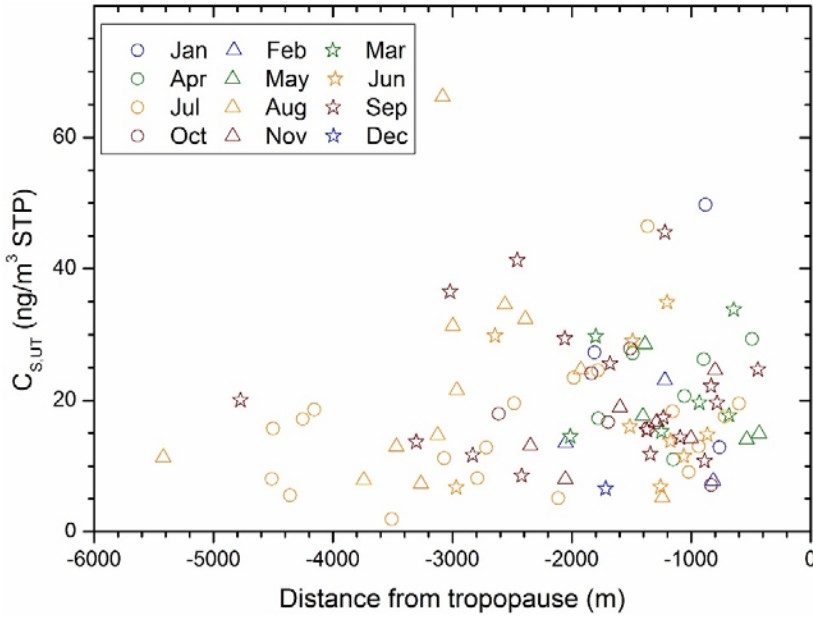

**Figure 1: Upper tropospheric particulate sulfur concentration related to distance from the tropopause, where the symbols indicate sampling month.**

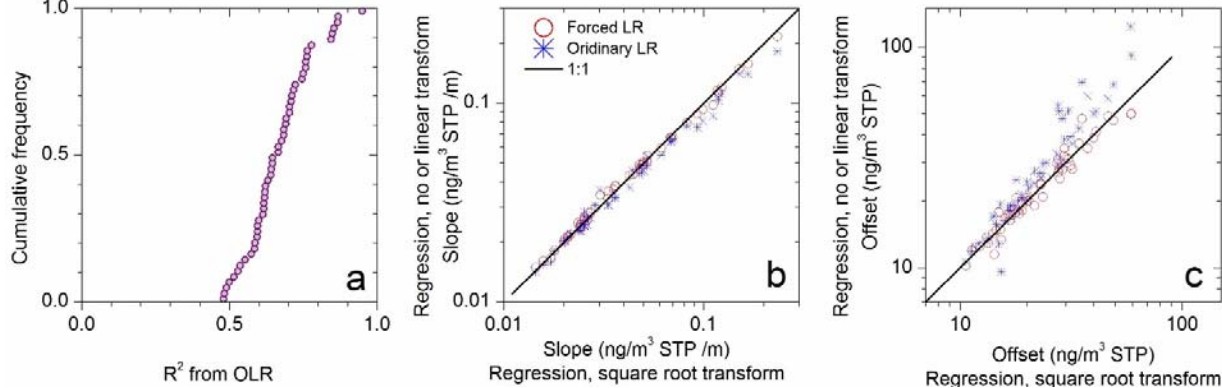

**Figure 2: a) Cumulative frequency of the coefficient of determination ($R^2$) of ordinary linear regression (OLR) between particulate sulfur concentration and distance from the tropopause of the 52 data groups used in this study. b) and c) Comparison of slope (Fig. 2b) and offset (c) between a square root transformed dependent variable and both OLR and forced linear regression.**

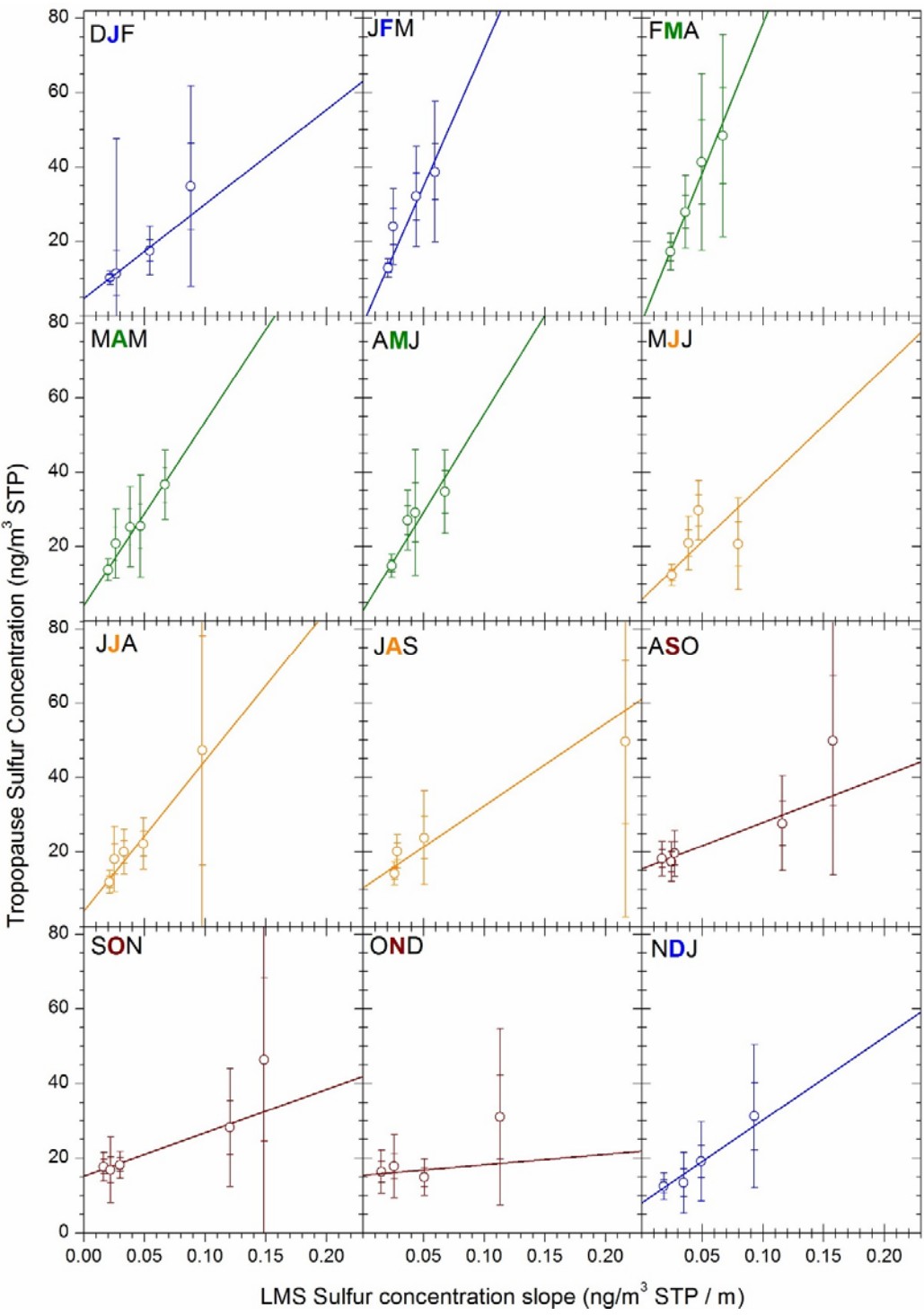

**Figure 3: Weighted regression between slopes and offsets of all groups of each season. The offset indicates the tropopause (and UT) concentration of tropospheric origin ($C_{S,UTtrop}$) and the slope of the fit (k) expresses the sensitivity of the UT concentration to changes in the stratospheric concentration slope (a). The error bars show and student-t 70% and 95% confidence interval, respectively.**

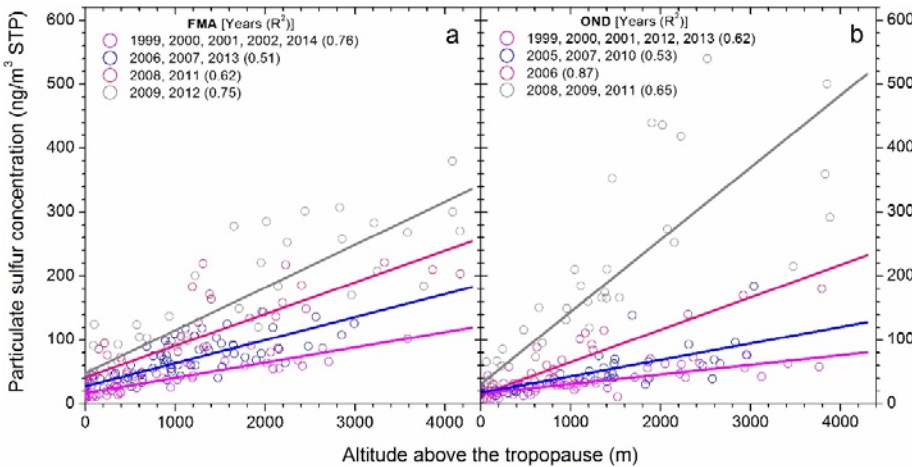

**Figure 4: Examples of first step forced linear regression between the particulate sulfur concentration and the altitude above the tropopause for the data groups of two seasons: a) FMA (February, March and April) and b) OND (October, November and December). The legend shows which years are included in the respective data groups followed by the coefficient of determination in paranthesis.**

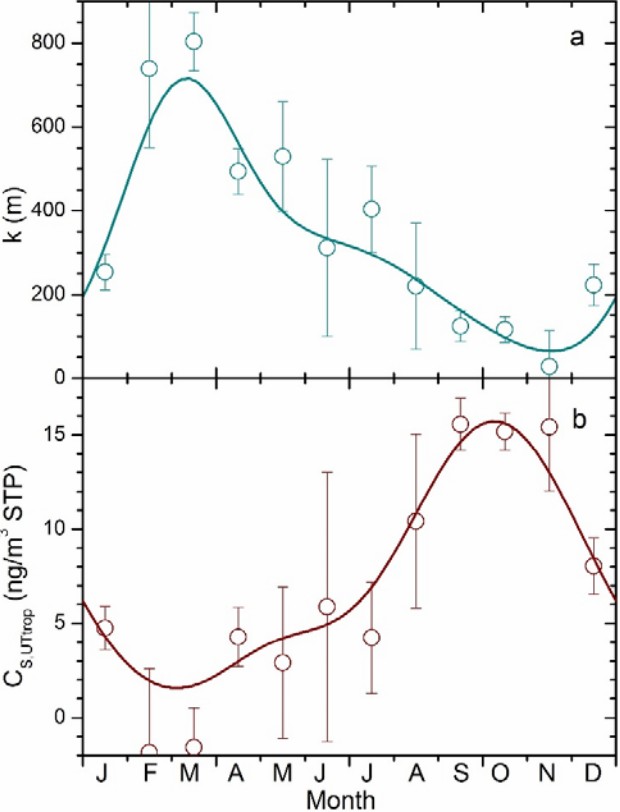

**Figure 5: Seasonal variation of (a) the sensitivity of the UT particulate sulfur concentration to the concentration slope in the LMS (k), and (b) the UT particulate sulfur concentration of tropospheric origin ($C_{S,UTtrop}$).**

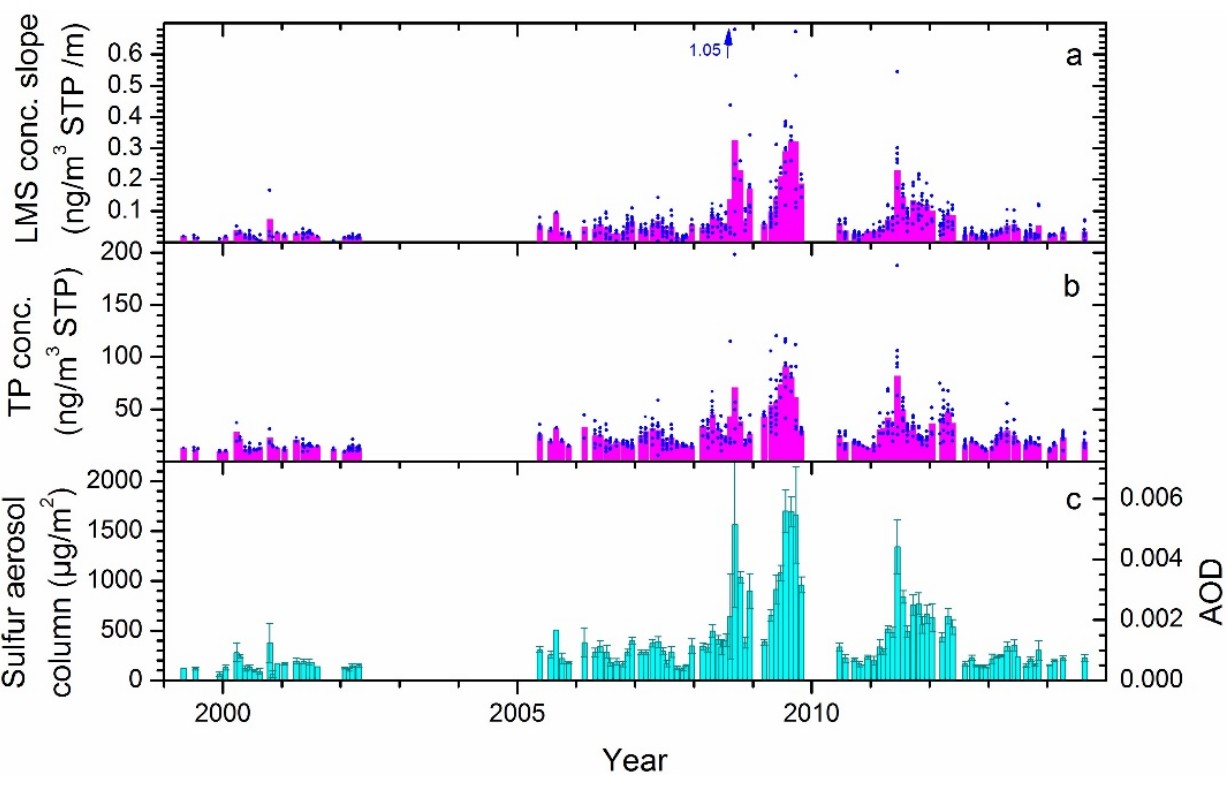

775

**Figure 6: Estimated a) slopes and b) offsets based on individual measurements (dots) and monthly averages (magenta bars). c) Sulfur aerosol column of the lowest 3000 m of the LMS with standard errors, assuming particles of 75% sulfuric acid and 25% water, and AOD (right y axis) assuming stratospheric background aerosol particle size distribution.**

780

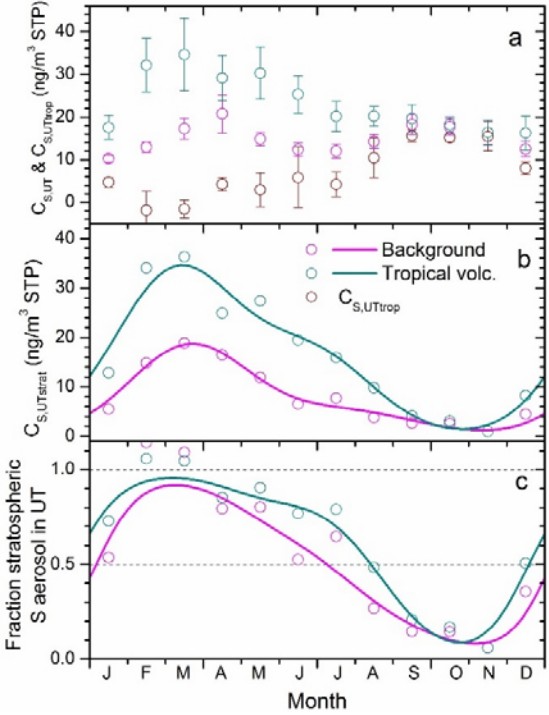

**Figure 7. Seasonal variation of a) the UT particulate sulfur concentration during LMS background conditions and moderate volcanic influence, together with the estimated UT particulate sulfur concentration of tropospheric origin ($C_{S,UTtrop}$), b) UT particulate sulfur concentration of stratospheric origin ($C_{S,UTstrat}$) obtained by subtracting the $C_{S,UTtrop}$ from the two UT concentration cases in Fig. 7a, and, c) the ratio $C_{S,UTstrat}$ /$C_{S,UT}$ for the two cases.**

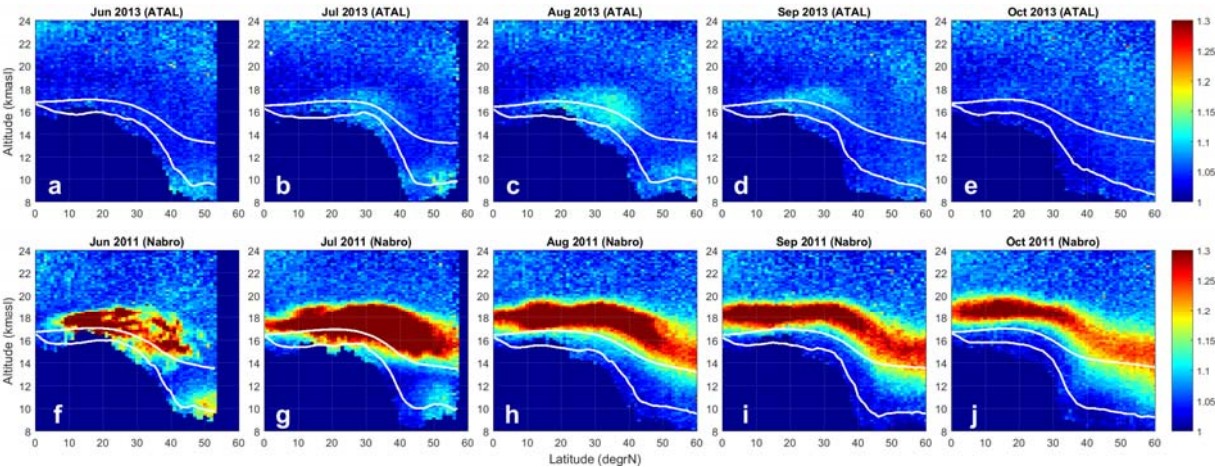

**Figure 8. Latitude and altitude distribution of scattering from aerosol in the stratosphere and the UT. Two series, from 2013 (upper row, a - e) and 2011 (lower row, f – j), show the monthly mean 532 nm wavelength scattering ratio from June to October without respectively with fresh volcanic influence. The former series allows identification of the comparably weak ATAL in July to September. The lower series includes two volcanic eruptions: the high altitude cloud from the tropical volcano Nabro (eruption in June 12, 2011) and the low altitude and midlatitude faint volcanic cloud from Grimsvötn (May 21, 2011). The two white lines in every graph show the monthly averaged positions of 380 K potential temperature (upper line) and the 1.5 PVU dynamical tropopause.**

