# Peer review of "Particulate sulfur in the upper troposphere and lowermost stratosphere - sources and climate forcing"

_Atmospheric Chemistry and Physics, 2016_

## Referee Comment (RC1) · Anonymous Referee #1 · 24 Feb 2017

This manuscript analyzes measurements of particulate sulfur taken from commercial aircraft altitudes in the IAGOS and CARIBIC programs. It attempts to derive the fraction of upper tropospheric sulfate that is from the stratosphere. This is an excellent data set but I find the analysis inadequate for several reasons. A previous paper by many of the same authors on a subset of these data (Friberg et al., 2014 but it only uses data through 2008) is a much better analysis. I would recommend resubmission with completely new analysis based on extending the Friberg et al. techniques to include the newer data and more emphasis than the Friberg paper on non-volcanic periods.

1) The manuscript (line 135) states that the analysis is based on concentrations per unit volume rather than mixing ratio. This is a mistake; straightforward analysis of fits

of concentration versus a parameter like distance from the tropopause requires the use of mixing ratio. For an idealized example, suppose the aircraft ascends 1 km as it goes through a perfectly uniform air mass. That ascent would change both the x-coordinate (distance from the tropopause) and the y-coordinate (concentration not corrected to mixing ratio). This coupling of the independent and dependent variables makes it impossible to interpret the slopes in a simple fashion. The stated reason for using a volume concentration (integration over an altitude range) can always be done later in the analysis.

2) The distance from a tropopause defined by potential vorticity (PV) is not a very good choice for the independent variable (Figures 1 and 2 and subsequent analysis). First, the PV values come from a meteorological analysis with substantial uncertainty. A colleague I spoke to estimated +/- 500 meters. There are also ambiguities with multiple tropopauses. Consider what Figure 2a would look like with uncertainties of +/- 500 m in the horizontal for most points, and more for a few points in the neighborhood of multiple tropopauses. Note that using an independent variable with significant uncertainty not only introduces noise into line fits but also biases the results to smaller slopes and, for positive data, larger intercepts. (This is worse than uncertainty in the dependent variable, which introduces noise but not bias.) Second, there is no reason why the gradient in PV has to be uniform with distance above the tropopause, so deeply stratospheric air could be close to or far above the tropopause. No tracer is perfect, but ozone, as used in Friberg et al., would be a far better choice than distance from a PV tropopause.

3) The analysis of the stratospheric influence fraction is very convoluted with no propagation of uncertainty shown. There are three successive line fits to data, as shown in Figure 2a to 2c. After reading through the manuscript several times, and having worked extensively with similar data, I still do not understand how the measurement uncertainties and atmospheric noise propagate into the results.

4) Putting aside the choice of independent and dependent variables and the propagation of uncertainty, there is a conceptual problem with defining the stratospheric influence from a corrected intercept derived from the line fits, as is done in this manuscript. Such an analysis of the slope and intercept of two variables in the lowermost mid-latitude stratosphere generally assumes that both variables are conserved quantities controlled largely by transport and mixing (e.g. Plumb 1996 JGR tropical pipe paper). But sulfate mass in the lowermost stratosphere is mostly controlled by sedimentation (Wilson et al., Steady state aerosol distributions..., ACP, 2008). In the presence of sedimentation, it is not obvious what the slope and intercept of a correlation plot mean. Indeed, it is clear from Figure 5 in Wilson et al. that a line fit over an altitude range that goes deep into the stratosphere could easily produce an intercept unrelated to the tropopause value.

Lesser concerns are: (a) Distance from the tropopause is strongly correlated with latitude, since commercial flights generally only get well above the tropopause at high latitudes. This makes it difficult to separate latitude and altitude as causes of a correlation. (b) The introduction is too broad.

---

## Referee Comment (RC2) · Anonymous Referee #2 · 22 Mar 2017

This manuscript reports on measurements of aerosol sulfur in aerosol samples collected from the IAGOS-CARIBIC platform over a 16 year period. Analysis focuses on a new regression technique that the authors suggest can be used to infer both the gradient of sulfur concentration and the integrated burden in the lowermost stratosphere (LMS), starting at the dynamical tropopause and extending to 3 km above it. It is also suggested that this analysis provides an estimate of the relative contribution of stratospheric sulfate mixed downward and tropospheric sources on the sulfur concentration in the upper troposphere (UT), and how these contributions vary seasonally.

My biggest problem with the paper is that the authors do not show and discuss enough raw data to demonstrate that the regression approach is reasonable. Figure 1 does

give the reader a useful impression of both the range and seasonality of sulfur observed in the UT, and supports the authors impression that there is little correlation with distance below the tropopause. However, the LMS concentration data are never shown so the reader has no idea if fitting linear regressions is a remotely logical approach. This is compounded by the fact that the analysis apparently required multiple steps which are not well explained in section 2.3.

I realize that this group has written a number of previous papers on this data set, and perhaps some of these have already presented spatial and seasonal distributions of sulfur in the LMS in ways that set the stage for this new analysis. However, I did not, and readers in general should not have to, read these earlier works to understand this one.

I could provide a fairly long list of specific sentences and paragraphs that I found to be confusing or misleading. However, I just noticed that reviewer 1 has suggested major revision, starting with fundamentally changing the approach to analysis, which will clearly require rewriting most of the text. Therefore it seems that specific editorial suggestions to improve clarity are premature

I agree with the concerns reviewer 1 raised regarding the use of concentrations rather than mixing ratios, and relying on distance from the tropopause as the independent indicator of degree of stratospheric character captured by a given sample. (I also note that simply defining this distance for sample intervals approaching 2 hours in length would often seem ambiguous, even without double tropopauses or crossings of the tropopause.). I cannot comment on the suggestion to extend the analysis used by Friberg et al., 2014, since, as noted above, I am not familiar with this paper.
* * *

---

## Referee Comment (RC3) · Anonymous Referee #3 · 28 Mar 2017

The paper describes the use of measurements of sulphur collected onto filters on board CARIBIC commercial aircraft over the past decade. The filters were analysed by PIXE. The data analysis develops a relationship between the concentration measurements and vertical distance to the tropopause derived from a PV analysis of re-analysis data. This is used to build a seasonal profile by combining data from multiple flights in each 3 monthly period. The analysis is used to show the amount of sulphur in the lower stratospheric column and derive an aerosol optical depth resulting from it.

I found the description of how the analysis was done to be less than clear. Some of the sentences were long and not transparent and on a number of occasions the sentence did to scan well or had typos in it e.g. "This was undertaken for 4 up to 7

groups of data for each season, and a total of 60 regression groups distributed over 12 overlapping seasons were used. This overlapping places each month in the center of a three-month season thus adjusting to smooth seasonal changes in the UTLS." I would recommend this whole section and how this relates to the further description of the approach in the results section need re-writing and clarifying. Further, it is not clear what is meant by "overlapping", does this mean that the some of the same data are used in multiple regressions? I assume this is a 3 month average centred on a particular month from the legend in figure 3. Please clarify. A previous referee is critical of the use of concentration rather than mixing ratio and I can see why in principle. Equally I can understand the authors' use of concentration since the column abundance of sulphur can be retrieved from the regressions and hence the AOD which would not be the case if the mixing ratio was used. This is also true of the use of altitude deriving from a PV definition of the tropopause rather than ozone. However, as far as I can see this only works if the pressure changes over the altitude range of the samples are sufficiently small that the regression derived results from the relative position to the tropopause and not the absolute altitude, this needs to be clarified before the analysis can be verified.

The weighting of the regression isn't described in sufficient detail for a reader to follow and replicate. This needs clarification.

---

## Author Comment (AC1) · 8 May 2017

Please see the supplement.

Please also note the supplement to this comment:
http://www.atmos-chem-phys-discuss.net/acp-2016-1142/acp-2016-1142-AC1-supplement.pdf

---

## Author Comment (AC3) · 8 May 2017

The comment was uploaded in the form of a supplement:
http://www.atmos-chem-phys-discuss.net/acp-2016-1142/acp-2016-1142-AC3-supplement.pdf

---

## Author Response (AR1)

**List of changes made followed by replies to the comments by the referees**

**Changes made:**

Table 1: No changes made

- Fig. 1: Similar to old version, but now based on mixing ratio
- *Fig. 2:* Removed but replaced by another fig that explains the methodology
- Fig. 3: The same as before except for that the change in methodology affected the details
- *Fig. 4:* A new fig that better explain the connection between the 2nd regression step and the measurement data
- *Fig. 5:* (old Fig. 4) The same as before except for that the change in methodology affected the details
- *Fig. 6:* (old Fig. 5) The same as before except for that the change in methodology affected the details
- *Fig.* 7: (old Fig. 6) The same as before except for that the change in methodology affected the details
- Fig. 8: New fig used to explore the ATAL

**Abstract**

- Added mention of CALIPSO, which is used in the new version
- Language improvements
- Adjusted the final 3 4 lines to comply with changes made to the manuscript

**1. Introduction**

- Added 4 lines about the ATAL
- Removed one sentence that repeats information given in the Discussion section
- Added 2.5 lines to further explain the need of chemical measurements to complement lidar measurements

**Methods 2.1**

- Added a line on aircrafts used and sampling time periods
- Exchanged 2 lines with 5 lines due to change from volume concentration to a mixing ratio.

**Methods 2.2**

- Changed two R2 values due to change from volume conc. to mixing ratio
- Added half a line to further explain the altitude parameter.
- Minor language adjustments

**Methods 2.3**

• Re-written the explanation of how the seasons overlap.

- Reduced number of data groups from 60 to 52 in order to avoid data groups containing few measurements.
- Moved the description of the data exclusions made (handling of outliers)
- Added approximately 1 page that illustrates and explains the choice of forced linear regression in the first regression step (between particulate sulfur concentration and altitude above the tropopause).
- Replaced old Fig. 2 with a new that shows the regression results based on different methods
- Simplified the regression by eliminating the use of ordinary linear regression (now only used for illustration purposes).
- Replaced approximately half a page describing the old methodology with approximately equally long text describing the new, including a clearer explanation of the error handling.
- Minor language adjustments

**Methods 2.4**

• New section. We make use of satellite data (CALIPSO) in the revised version of the manuscript in order to further explain the results obtained. This section gives the experimental details.

**3. Results**

- Made some minor changes in the first section due to the changes in methodology
- Added 1/4 page to comment a new (added) Fig. 4 that shows the connection between the two regression steps used.
- Old equation 9, now 7, was changed to accommodate the change from volume concentration to mixing ratio in the regressions, and ¼ page was added to explain how the column concentration can be obtained from a mixing ratio.
- Minor language adjustments

**4. Discussion**

- The change in the regression methodology did somewhat change the seasonal dependence of the tropospheric source of UT particulate sulfur (Fig. 5b; old Fig 4b) that induced some small changes to the wording in the two first sub-sections.
- Descriptions of extratropical transport of boundary layer air to the UT has been reduced from 1/2 to 1/4 and moved approximately 1 page up.
- The discussion of the seasonal dependence of the tropospheric source of the UT particulate sulfur has been changed by simplifying the descriptions of relations to other species.
- The new look of the seasonal dependence of the tropospheric source of the UT particulate sulfur with a very clear peak in the fall (Fig. 5b in the new version) prompted further analysis where we found that the Asian monsoon, which forms an aerosol layer at 14 18 km altitude (ATAL) was the most likely cause. Approximately 1 page was added and a Fig. (8) was added containing 10 images obtained from the CALIOP sensor of CALIPSO.

**5. Conclusions**

• 4 lines of text was replaced by 10 lines to accommodate the new findings, mainly on the ATAL.

**Reply to referees**

Referees#2 and #3 made comments on the review of referee#1. We therefore include all answers in this file (referee comments in blue). We would also like to apologize to Referees#2 and #3 for making frequent references to the answer to Referee#1, because we first answered the comments of the latter.

**Summary:**

We thank the three referees for the comments. In response we have adjusted the data handling by using mixing ratios which are converted to volume concentrations in a late stage of the evaluation. We also have simplified and clearer motivated the choice of regression methodology in a new Fig. 2 (b and c), as well as clearer explained the weighting used in the second regression step. We have re-written section 2 accordingly. The changes did of course affect all Figs., and in particular the tropospheric source of particulate sulfur where a clear fall maximum was obtained. In order to explain this feature, further analyses were undertaken based on lidar data from the CALIPSO satellite. We found that the aerosol layer ATAL, formed from the Asian monsoon at 14 - 18 km altitude, is the likely cause. This added one fig (Fig. 8). We have also improved the presentation of the original data by adding a cumulative frequency distribution of the coefficient of determination (Fig. 2a), as well as adding a new fig (Fig 4) that clearer shows the first step regression and hopefully increase the understanding of the relations between the two regression steps used. We have also changed the description of the seasons used, and in general tried to improve the legibility.

**Answers to Referee#1**
This manuscript analyzes measurements of particulate sulfur taken from commercial aircraft altitudes in the IAGOS and CARIBIC programs. It attempts to derive the fraction of upper tropospheric sulfate that is from the stratosphere. This is an excellent data set but I find the analysis inadequate for several reasons. A previous paper by many of the same authors on a subset of these data (Friberg et al., 2014 but it only uses data through 2008) is a much better analysis. I would recommend resubmission with completely new analysis based on extending the Friberg et al. techniques to include the newer data and more emphasis than the Friberg paper on non-volcanic periods.

We appreciate the kind words about Friberg et al. (2014). The entire author group is well acquainted with the usefulness of O3 in interpreting particulate sulfur in the LMS. More than 10 years ago we developed an O3-based model to study particulate sulfur in the volcanically quiescent period at the turn of the

millennium (Martinsson et al., JGR 2005). Following the Kasatochi eruption we introduced the ratio of particulate sulfur to O3 in order to express volcanic influence on the concentration of the former species (Martinsson et al., GRL 2009), which make samples that were variably impacted by mixing across the tropopause more comparable. In the present study we seek a standalone methodology to be able to express concentration gradients and to integrate to obtain radiative impact. This is important because this data allow us to identify the volcanic component also close to the tropopause in contrast to lidar measurements that are biased by tropospheric sources (dust, condensed water and other species). This is altogether different in the goals compared to our previous studies, and to our knowledge the first such study. Therefore, a different methodology is required.

1) The manuscript (line 135) states that the analysis is based on concentrations per unit volume rather than mixing ratio. This is a mistake; straightforward analysis of fits of concentration versus a parameter like distance from the tropopause requires the use of mixing ratio. For an idealized example, suppose the aircraft ascends 1 km as it goes through a perfectly uniform air mass. That ascent would change both the x-coordinate (distance from the tropopause) and the y-coordinate (concentration not corrected to mixing ratio). This coupling of the independent and dependent variables makes it impossible to interpret the slopes in a simple fashion. The stated reason for using a volume concentration (integration over an altitude range) can always be done later in the analysis.

This problem was considered already before the analysis of the data, but was deemed insignificant due to the relatively narrow altitude range of the measurements, which, in principle, was a mistake. In response to this criticism we did make a change in the analysis to undertake the regression analyses based on STP-normalized concentrations ( $C_{STP} = aZ + b$ ), which is a mixing ratio. This means that the regression results need to be converted to volume concentration in the final integration ( $C_{STP}/C_V = Q_V/Q_{STP}$ , where C and Q are concentrations and molar volumes). We used ECMWF data to estimate the slope of the molar volume from pressure and temperature profiles in the LMS. Based on the average slope of all our samples we describe the molar volume with an exponential function:

 $Q(Z) = Q_{TP} e^{wZ}$ , where Z is the height above the TP and w = 0.0001535 m-1 (from ECMWF).

For each measurement we have a molar volume  $Q_m$  obtained at height  $Z_m$  above the TP. For this measurement the TP molar volume is obtained by  $Q_{TP} = Q_m e^{-wZm}$ . We integrate to obtain the column concentration for the first 3000 m of the LMS:

$$C_{col} = \int_0^{3000} \frac{Q_{STP}}{Q_V(Z)} C_{STP}(Z) dZ = \int_0^{3000} \frac{Q_{STP}}{Q_{TP}} e^{-wZ} (aZ+b) dZ$$

2) The distance from a tropopause defined by potential vorticity (PV) is not a very good choice for the independent variable (Figures 1 and 2 and subsequent analysis). First, the PV values come from a meteorological analysis with substantial uncertainty. A colleague I spoke to estimated +/- 500 meters. There are also ambiguities with multiple tropopauses. Consider what Figure 2a would look like with uncertainties of +/- 500 m in the horizontal for most points, and more for a few points in the neighborhood of multiple tropopauses. Note that using an independent variable with significant uncertainty not only introduces noise into line fits but also biases the results to smaller slopes and, for positive data, larger intercepts. (This is worse than uncertainty in the dependent variable, which introduces noise but not bias.) Second, there is no reason why the gradient in PV has to be uniform with distance above the tropopause,

so deeply stratospheric air could be close to or far above the tropopause. No tracer is perfect, but ozone, as used in Friberg et al., would be a far better choice than distance from a PV tropopause.

Locally there can be large uncertainties in the model tropopause. However, a sample was taken over typically 1500 km or more in flight range, meaning that the distance to the tropopause is based on typically 100 ECMWF positions along the flight track. Therefore, these samples become less sensitive to local errors of the model. The referee need to acknowledge the different purpose here compared to other studies of the ExTL. The objection "there is no reason why the gradient in PV has to be uniform with distance above the tropopause" is difficult to interpret. We do not use the PV as a tracer, and hence we do not need to make any assumptions concerning PV uniformity. Instead we use the dynamical TP at 1.5 PVU as the lower boundary of the stratosphere, which is generally recognized as the lower boundary of the stratosphere in view of chemical composition. Then we use the linear distance to this TP in the regression analyses, which tests the uniformity. The coefficient of determination (R2) exceeds 0.6 in 72% of the 52 ordinary linear regressions undertaken. The deviations from the linear models consists of scatter without any trends. We do not force an erroneous shape of the gradient.

We considered using an O3-based distance according to Sprung and Zahn (JGR 2010) which is available in the IAGOS-CARIBIC dataset. However, we decided against that for two reasons:

- A) That parameter refers to a static tropopause implying that approximately 1000 m of the LMS falls below.
- B) The use of a tracer would potentially introduce non-linearity in the vertical parameter that could introduce a bias in the results

3) The analysis of the stratospheric influence fraction is very convoluted with no propagation of uncertainty shown. There are three successive line fits to data, as shown in Figure 2a to 2c. After reading through the manuscript several times, and having worked extensively with similar data, I still do not understand how the measurement uncertainties and atmospheric noise propagate into the results.

Due to the good detection capacity for particulate sulfur, the total measurement errors are 12% (Martinsson et al., AMT 2014), which is small compared with atmospheric variation of this species. In the analysis we use a method of forcing the linear regression to comply with the observations close to the TP, as a way to deal with the heteroscedastic nature of the data. In response to this comment we have simplified the analysis. We skipped the ordinary linear regression (now only used for illustration purpose) and used only forced linear regression, where the regression was forced to the average concentration  $C_0$  and distance  $Z_0$  of the measurements closest to the tropopause. With a simple linear variable transformation (forming  $C - C_0$  and  $Z - Z_0$ ) this becomes equivalent to forcing the regression through the origin. Thus, first a forced linear regression, and then a weighted regression between the obtained slopes and offsets of each month. The uncertainty in the slopes of the first step regression is propagated. The error estimates used in Fig 3 are based on the student t distribution (t70% and t95%, two-sided) because the number of measurements available to estimate  $C_0$  in some cases is small. The error in b (at Z = 0) is based on the uncertainties in  $C_0$  (at  $Z_0$ ) and the slope, and is computed using the weakest slope at the upper error limit of  $C_0$  and strongest slope to the lower limit.

To further advocate this method, we undertook a small study of the scedastic nature of our data using variable transformations. The most common transforms, square root and logarithm of the dependent variable, were investigated. Like most natural science data set our is heteroscedastic, showing increasing

variance with distance from the TP. The log transformation changed the data in a way that the variance became smallest at large distance from the TP, whereas the square root transformation resulted in a rather constant variance in the dependent variable along the independent variable axis. This transform is thus more suitable for regression. After transformation the relation between the dependent (y' = sqrt(y)) and the independent (x) variable is not linear (y and x has a linear relation). Thus the following expression should be minimized with respect to a and b:  $(y' - sqrt(ax + b))^2$ . This resulted in rather tedious expressions that we had to solve numerically. In the Fig (Fig. 2 in a revised manuscript) you can see a comparison of slope (Fig b) and offset (Fig c) between the sqrt-transformed regression and ordinary linear regression (OLR) and forced LR. As can be seen, the OLR deviates sometimes strongly, in particular the offset, from the sqrt-transformed due to the heteroscedastic data. The forced LR, on the other hand, shows only small differences from the analysis based on sqrt-transformation. Due to more direct determination of the offset as well as the simpler analysis we chose to use forced LR.

4) Putting aside the choice of independent and dependent variables and the propagation of uncertainty, there is a conceptual problem with defining the stratospheric influence from a corrected intercept derived from the line fits, as is done in this manuscript. Such an analysis of the slope and intercept of two variables in the lowermost midlatitude stratosphere generally assumes that both variables are conserved quantities controlled largely by transport and mixing (e.g. Plumb 1996 JGR tropical pipe paper). But sulfate mass in the lowermost stratosphere is mostly controlled by sedimentation (Wilson et al., Steady state aerosol distributions. . ., ACP, 2008). In the presence of sedimentation, it is not obvious what the slope and intercept of a correlation plot mean. Indeed, it is clear from Figure 5 in Wilson et al. that a line fit over an altitude range that goes deep into the stratosphere could easily produce an intercept unrelated to the tropopause value.

Yes, there is sedimentation going on in the stratosphere, which, through the Cunningham slip correction factor describing the viscous contact between a particle and the surrounding air, is strongly dependent on altitude. In the LMS, with typical size distributions (background or moderately volcanically influenced), the sedimentation velocity is typically less than 0.15 km/month, implying that the particulate sulfur mainly is removed from the LMS by air transport (Martinsson et al., JGR 2005). Considering the recommendation above by the referee to use a tracer to improve the analysis, this comment is somewhat surprising. Whereas this effect in principle could affect the relation to a tracer, the observed gradient in our present study carries no such assumptions.

Lesser concerns are: (a) Distance from the tropopause is strongly correlated with latitude, since commercial flights generally only get well above the tropopause at high latitudes. This makes it difficult to separate latitude and altitude as causes of a correlation. (b) The introduction is too broad.

The referee is right in the assumption that observations deep into the LMS could be more frequent at higher latitudes. Ideally we would want to have enough observations to also study the latitude dependence. However, the salient features of the LMS with its ExTL are caught by the broad latitude band.

Answer to referee#2 follows

**Answers to Referee#2**
This manuscript reports on measurements of aerosol sulfur in aerosol samples collected from the IAGOS-CARIBIC platform over a 16 year period. Analysis focuses on a new regression technique that the authors suggest can be used to infer both the gradient of sulfur concentration and the integrated burden in the lowermost stratosphere (LMS), starting at the dynamical tropopause and extending to 3 km above it. It is also suggested that this analysis provides an estimate of the relative contribution of stratospheric sulfate mixed downward and tropospheric sources on the sulfur concentration in the upper troposphere (UT), and how these contributions vary seasonally.

My biggest problem with the paper is that the authors do not show and discuss enough raw data to demonstrate that the regression approach is reasonable. Figure 1 does C1 give the reader a useful impression of both the range and seasonality of sulfur observed in the UT, and supports the authors impression that there is little correlation with distance below the tropopause. However, the LMS concentration data are never shown so the reader has no idea if fitting linear regressions is a remotely logical approach.

The reason we have not included figs of the first step linear regressions is that they are so many. In Fig 3 you in principle can see them all, but we agree that it is difficult to get the impression of the data from that Fig. However, a useful compromise could be to, instead of making 52 graphs, present the R2 of the 52 ordinary linear regressions (OLR). Fig. 2a (the new version) shows the cumulative frequency of R2. As can be seen, R2 of the 52 OLRs span 0.48 to 0.95 and 50% of the OLRs have R2 exceeding 0.66. We have also added a new fig (Fig 4) that together with Fig 3 better explain how the actual measurements are connected with the regression in the second step, which in Fig. 5 becomes the seasonal dependences of k(s) and Cs,UTtrop(s), which in turn are used to obtain ai and bi in Figs. 6a and b.

**This is compounded by the fact that the analysis apparently required multiple steps which are not well explained in section 2.3.**

As explained in the answer to Referee#1 we have simplified the data evaluation, please see above for details.

I realize that this group has written a number of previous papers on this data set, and perhaps some of these have already presented spatial and seasonal distributions of sulfur in the LMS in ways that set the stage for this new analysis. However, I did not, and readers in general should not have to, read these earlier works to understand this one. I could provide a fairly long list of specific sentences and paragraphs that I found to be confusing or misleading. That would have been helpful.

However, I just noticed that reviewer 1 has suggested major revision, starting with fundamentally changing the approach to analysis, which will clearly require rewriting most of the text.

Based on the comments from all the referees the revision should mainly concern the methods section. The revision of the methods has, however, induced changes in other places, mainly the discussion section.

Therefore it seems that specific editorial suggestions to improve clarity are premature I agree with the concerns reviewer 1 raised regarding the use of concentrations rather than mixing ratios, and relying on distance from the tropopause as the independent indicator of degree of stratospheric character captured by a given sample. (I also note that simply defining this distance for sample intervals approaching 2 hours in length would often seem ambiguous, even without double tropopauses or crossings of the tropopause.). I cannot comment on the suggestion to extend the analysis used by Friberg et al., 2014, since, as noted above, I am not familiar with this paper.

We have made a new analysis based on mixing ratios, and extensively answered the comments by referee#1, please see above.

Answer to referee#3 follows

**Answers to Referee#3**
The paper describes the use of measurements of sulphur collected onto filters on board CARIBIC commercial aircraft over the past decade. The filters were analysed by PIXE. The data analysis develops a relationship between the concentration measurements and vertical distance to the tropopause derived from a PV analysis of re-analysis data. This is used to build a seasonal profile by combining data from multiple flights in each 3 monthly period. The analysis is used to show the amount of sulphur in the lower stratospheric column and derive an aerosol optical depth resulting from it.

I found the description of how the analysis was done to be less than clear. Some of the sentences were long and not transparent and on a number of occasions the sentence did to scan well or had typos in it e.g. "This was undertaken for 4 up to 7 groups of data for each season, and a total of 60 regression groups distributed over 12 overlapping seasons were used. This overlapping places each month in the center of a three-month season thus adjusting to smooth seasonal changes in the UTLS." I would recommend this whole section and how this relates to the further description of the approach in the results section need re-writing and clarifying.

We have looked carefully on the formulations. We have also simplified the analysis (see answer to referee#1), which hopefully will further help to improve legibility.

Further, it is not clear what is meant by "overlapping", does this mean that the some of the same data are used in multiple regressions? I assume this is a 3 month average centred on a particular month from the legend in figure 3. Please clarify.

Your interpretation is correct. We have clarified this in the revised manuscript.

A previous referee is critical of the use of concentration rather than mixing ratio and I can see why in principle. Equally I can understand the authors' use of concentration since the column abundance of sulphur can be retrieved from the regressions and hence the AOD which would not be the case if the mixing ratio was used. This is also true of the use of altitude deriving from a PV definition of the tropopause rather than ozone. However, as far as I can see this only works if the pressure changes over the altitude range of the samples are sufficiently small that the regression derived results from the relative position to the tropopause and not the absolute altitude, this needs to be clarified before the analysis can be verified. We have changed the analysis in this respect to follow the recommendation of Referee#1 of using mixing ratios, and complemented with the use of a transformation to volume concentrations in the integration of the column concentrations.

**The weighting of the regression isn't described in sufficient detail for a reader to follow and replicate. This needs clarification.**

The weighting is undertaken based on the error estimated of each forced LR (in step 1), where the inverted, squared linear error (t70%; see answer to Referee#1) of each first step regression is used as weights in the second step regression. This is clarified in the revised manuscript.

**Particulate sulfur in the upper troposphere and lowermost stratosphere - sources and climate forcing**

Bengt G. Martinsson1, Johan Friberg1, Oscar S. Sandvik1, Markus Hermann2, Peter F.J. van Velthoven3 and Andreas Zahn4

1Division of Nuclear Physics, Lund University, Sweden
 2Leibniz Institute for Tropospheric Research, Leipzig, Germany

[revised manuscript text omitted]
 $(\underline{R}_{2}^{2})$ of the 52 OLRs undertaken. $\underline{R}_{2}^{2}$ spans 0.48 to 0.95, 72% of the groups having $\underline{R}_{2}^{2}$ exceeding 0.6 |        | Formatt |
|     | and 50% exceeding 0.66. The deviations from the OLR models consists of scatter that does not show any trends.                                                                 | $\sim$ | Formati |
|     | When investigating the variance of the dependent variable (the particulate sulfur concentration, Cs) along the                                                                |        | Formati |
|     | independent variable (altitude above the tropopause, Z) it is clear that the variance of $C_{\underline{S}}$ increases with increasing Z.                                     |        | Formati |
| 225 | This heteroscedastic nature of the data, which is shared by most natural science data sets, could unfavorably affect in                                                       |        | Formati |
|     | particular the offset of the OLR. In order to further investigate the effects of this problem two variable                                                                    |        |         |
|     | transformations of the dependent variable were tested: logarithmic and square root transformations. The logarithmic                                                           |        |         |
|     | transformation turned the problem around to the other side, i.e. the logarithm of Cs has large variance for small Z and                                                       |        | Formati |
|     | small variance for large Z. The square root transformation of C s , on the other hand, shows rather constant variance                                              |        | Format  |
| 230 | along the Z axis, thus making this transformation more suitable for regression. The transformation $C'_{S} = \sqrt{C_{S}}$ was                                                |        |         |
|     | applied to the data. As already pointed out, $C_{\delta}$ and Z has a linear relation implying that the following expression                                                  |        | Formatt |
|     | should be minimized with respect to slope (a) and offset (b): $(C'_S - \sqrt{aZ + b})^2$ . This results in rather tedious                                                     |        |         |
|     | expressions that we solved numerically for all the 52 data groups. Figures 2b and c show comparisons between the                                                              |        |         |
|     | slopes and the offsets obtained by the square root transformed regression results and the OLRs. It is clear that the                                                          |        |         |
| 235 | heteroscedastic nature of the data causes large deviations, especially in the offset.                                                                                         |        |         |
|     | Vet another transformation, based on forcing the regression to comply with the data for small 7, was investigated. To                                                         | /      | Format  |
|     | that end, the average concentration ( $C_{8,0}$ ) of the data points closest to the tropopause at the average distance $Z_0$ from                                             |        | Format  |
|     | the tropopause was formed. $C_{80}$ of the 52 data groups is on average based on 13 measurements, and the minimum                                                             |        | Formati |
|     | was 5, and the average $Z_0$ is 88 m and the largest is 273 m. The data were transformed linearly to place $C_{80}$ and $Z_0$ in                                              |        | Format  |
| 240 | the origin by forming $C_{\delta'} = C_{\delta 0}$ and $Z' = Z - Z_0$ followed by linear regression forced through the origin, i.e. $C_{\delta'}$                             |        | Formati |
|     | = aZ'. Finally, the regression results are transformed back to the form $C_{\delta} = aZ + b$ , where the slope a is not changed                                              |        | Formati |
|     | by the translation, and the offset is obtained by $b = C_{8.0} - aZ_0$ . These results of the forced linear regressions are                                                   | M      | Formati |
|     | compared with the square root transformed regressions in Figures 2b and c. As can be seen, the forced linear                                                                  |        | Formati |
| 245 | regressions, in contrast to the OLRs, show only small deviations from the square root transformed data. Due to the                                                            |        | Formati |
|     | more direct determination of the offset as well as the simplicity of forced linear regression compared with square                                                            |        | Formati |
|     | root transformation, the forced linear regression method is chosen for the analyses. Thus, for each season (s) and year                                                       |        | Formati |
|     | (y) in total 52 <del>60</del> forced linear regressions were undertaken:                                                                                               | //     | Formati |
|     |                                                                                                                                                                               |        | Format  |

 $\mathcal{C}_S(y,s,Z) = a(y,s)Z + b(y,s)$

where a and b varies with season and the strength of the volcanic influence.

(2)

[revised manuscript text omitted]

The estimated slopes and offsets are shown in Figs. 5a-6a and b, where the dots are individual measurements and the histogram monthly averages. Here measurements taken at an altitude of less than 50 m above the tropopause are not shown, because they were judged to have too small stratospheric character for an estimate of the concentration slope in the LMS. Both the slope and the tropopause concentration are affected by volcanism (Table 1), but the relative response of the slope is much stronger than that of the tropopause concentration, see e.g. the falls of 2008 and 2009 affected by the Kasatochi and Sarychev eruptions.

Observations at various altitudes above the tropopause are difficult to compare, due to the concentration gradient in the LMS. With the estimates of the tropopause concentration and the slope in the particulate sulfur LMS concentration, each measurement becomes an estimate of the total amount of particulate sulfur in the altitude interval investigated by integration of eq. 64.÷ However, first the STP concentrations (C8.STP = C8 = aZ + b) need to be converted to volume concentrations (C8.V), which are related by C8.V = C8.STP Q8.TP/QV(Z), where Q are molar volumes. For that purpose, the altitude dependence of the molar volume from the tropopause up to 5 km above the tropopause was extracted from temperatures and pressures obtained from ECMWF for each sample. The molar volume can be expressed as QV(Z) = QV(0)ewZ, where the tropopause is at Z = 0 and w = 0.0001535 m2-1 obtained as the average of all samples. For a measurement the molar volume is Qm obtained at distance Zm from the tropopause, and the tropopause molar volume is computed by QV(0) = Qme-wZm. Finally, the column concentration of particulate sulfur for the first 3 km above the dynamical tropopause of 1.5 PVU is obtained by integration of the volume concentration;

$$\underline{C}_{S,col} = \int_0^{3000} \frac{Q_{STP}}{Q_V(Z)} C_{S,STP}(Z) dZ = \int_0^{3000} \frac{Q_{STP}}{Q_V(0)} e^{-wZ} (aZ+b) dZ \underline{$$

 $\frac{\Phi}{(a''(y,s)Z + a''(y,s)k(s) + C_{\underline{s},\underline{UTtrop}}(s))}dZ$

|   | Formatted: Line spacing: 1.5 lines |
|---|------------------------------------|
| - | Formatted: Font: (Default) Times   |
|   | New Roman, 10 pt                   |

(7)

[revised manuscript text omitted]

**820 Tables**

**Table 1.** Most sig

---

## Referee Report (RR1)

Review of Paper: **Particulate sulfur in the upper troposphere and lowermost stratosphere – sources and climate forcing** by Martinsson et al. 2017, submitted to *Atmospheric Chemistry and Physics*

Martinsson et al. studied sources regions of upper tropospheric sulfur using long-term IAGOS-CARIBIC data. The authors show that the relative contributions (either from transport below or mixed down from the stratosphere) in UT sulfur are dependent on seasons, volcanic activities. The paper has gone through the first round of review with substantial changes based on previous reviewers especially reviewer #1. In general the paper is in good shape, but some concerns (especially on tropospheric contributions) need to be addressed before publication.

a. In the abstract, please give more definition on UT and LMS, i.e. LMS is ~3 km above dynamic tropopause; while UT is how many km below?

b. Line 25 *"We find that tropospheric sources dominate during summer andthe fall as a result of downward transport of the Asian tropopause aerosol layer (ATAL) formed in the Asian monsoon"*
Line 576 *"The ATAL is transported downwards, and affects the extratropical tropopause region in August to December."*
These arguments are unclear to me, what does "downwards transport" mean? A recent paper "efficient transport of tropospheric aerosol into the stratosphere via the asian summer monsoon anticyclone" shows that ATAL is mainly lifted above locally to the stratosphere and further transport to high latitudes. See their Figure 4.

c. The same paper above argues that in summer/fall seasons, convections transport tracer gases (SO2) and aerosol to UT. Do you think it maybe more likely a reason for higher tropospheric contribution of UT sulfur rather than downward transport of ATAL?

d. *"SO2 measurements by the satellite-based instrument MIPAS indicate a UT seasonal variation in the NH midlatitudes with low concentrations in December to March and the highest concentrations in June to September (Höpfner et al., 2015)."*
In my impression MIPAS SO2 product is of large uncertainties, and may not be meaningful in UT. Shown in Rollins et al. (2017, GRL), MIPAS overestimates the SO2 concentration in TP by a factor of ~3 or so.

e. I found the discussions on SO2, CO, OH are difficult to understand. Seems authors claim that direct transport of SO2 and sulfate aerosol from lower troposphere is not going to explain the seasonal variation on tropospheric contribution to UT sulfur, because they will be washed out or cloud-processed. And the authors suggest the concentrations of OH and SO2 are responsible. Well, are we certain that new TP, SO2 oxidation is OH-limited? Any know seasonal cycle on SO2 emissions can explain the seasonal variation? I think some surface SO2 can survive in UT especially in deep convection.

---

## Author Response (AR2)

Answers to Referees. We would like to thank you for your efforts to improve our manuscript by your comments. Answers from the authors in blue text. Line references of the authors refer to manuscript prior to the present revision.

**Referee 2:**

This manuscript reports on measurements of aerosol sulfur in aerosol samples collected from the IAGOS-CARIBIC platform over a 16 year period. Analysis focuses on a new regression technique that the authors suggest can be used to infer both the gradient of sulfur concentration and the integrated burden in the lowermost stratosphere (LMS), starting at the dynamical tropopause and extending to 3 km above it. It is also suggested that this analysis provides an estimate of the relative contribution of stratospheric sulfate mixed downward and tropospheric sources on the sulfur concentration in the upper troposphere (UT), and how these contributions vary seasonally. Compared to the original version, changes in the data processing now produce more pronounced seasonality in the estimated contribution of S derived from tropospheric sources to the S burden in the UT. In particular, the analysis now suggests a quite strong peak which authors attempt to link to downward transport from the ATAL.

While the approach has changed in response to comments on first version, and some sections describing the data analysis are now more clear, I still find myself not very convinced. Step 1, doing forced linear regression (FLR) of S versus distance above the tropopause is relatively straight forward, except for one key point. The data "were grouped with respect to concentration levels of the different years, resulting in 4 to 5 groups of data for each season" (lines 179-180). The discussion about this step seems to justify at least 2 groups (background and influenced by volcanoes), and perhaps the volcanic group could be (was?) further divided into strong and weak influence. It seems that the authors recognized that there had to be more than 2 groups in each season to do step 2 regressions, but how these were created is not explained. Some objective method of creating these groups, perhaps based on mean, median, max measured S concentrations, or maybe based on time since eruptions were known to have perturbed the LMS burden (based on previous work by this group using largely the same data set) would seem essential, especially for anyone attempting to replicate the analysis or apply the technique to some future similar data set. As it reads now, it almost seems that the approach was to sort by year based on timing of volcanos, but then move years (or individual samples?) that didn't work into another group until things looked better

It would be difficult to use a priori knowledge to classify the effects of a given volcano. We still miss accurate inputs to global models that describe the distribution and the evolution of volcanic injections to the stratosphere (http://volmip.org/). We did not have one volcanic group because the eruptions vary in strength and distribution. Fig.4 shows two examples where you can see that the "background" and "moderate tropical volcanism" groups are closely spaced, but still clearly separated, see e g Friberg et al in Tellus 2014. The need for further groups of data is obvious (Fig. 4) to cover the strong influences of e.g. Kasatochi, Sarychev and Nabro.

To clarify, we have added (line 183): "The remaining data of each year were tested for systematical differences. Those years where the data overlapped in particulate sulfur – height above the tropopause space, were grouped together."

The next step in the analysis is now basically clear conceptually: for each season the intercept and slope from 4 or 5 FLR in step 1 were regressed against each other, using weighted regression with the weighting based on estimated uncertainty in the FLR fits. The explanation of this weighting is not completely clear to me, but seems to generate uncertainty only in the FLR intercept (termed offset) when transferred to Fig 3 which shows results of the weighted regressions by season. Surely, the slopes from step 1 were also uncertain. It would seem that uncertainty in both slope and intercept from the FLR would propogate into uncertainty in the estimated slope and intercept from the weighted regression. Based on the small number of data points and the magnitude of the error bars in all 12 panels of Fig 3, I find the small error bars in Fig 5 very surprising (seems that error bars in 5 a should come directly from uncertainty in the slope from fits in Fig 3, while those in 5 b come from uncertainty in the intercepts in Fig 3).

Sorry but we do not understand this comment. Fig. 5a collects the slopes obtained from the regressions shown in Fig. 3, and Fig. 5b collects the intercepts from the same figure. The errors of the two quantities are computed and presented in Fig. 5. This is clearly explained in the manuscript.

In particular, the authors need to carefully consider how to propagate uncertainty from both initial measurement, uncertainty in Z, and uncertainty in slope and intercept of the FLR to the intercept values for the weighted regressions in Fig 3 and clearly explain how they do this, since they assert that the derived values represent the concentration of S in the UT that is derived from tropospheric sources and use these numbers for most of the remaining analyses in the paper. Lines 244 - 249 address how they attempted to account for the uncertainty from FLR, but not in very clear manner.

Of course the errors from step 1 very much affects the results in step 2. Imagine that we substantially increase the errors in step 1. The first visible sign would be that the data points would be more scattered in step 2, because the errors are larger. That would affect the step 2 regression and induce larger errors in the estimated  $C_{S,UTtrop}$  and k. The second way the results of step 1 enters step 2 is by using the errors from step 1 as weights, where data points with small variance affects the regression more than those with large variance. If you are still not convinced, try to draw alternative fits in fig 3, minding the size of the error bars, and you will see that the errors given for  $C_{S,UTtrop}$  and k are highly realistic.

The referee is right in that the slope estimates also have uncertainties. The average variance ratio was 0.19, i.e. the relative variance of the slope divided with that of the intercept from the step 1 FLR. The slope errors thus are small compared to those of the intercept, and were therefore neglected.

On line 237 the following was added: The relative variance of the slopes was much smaller than that of the offsets (average variance ratio of 0.19). For simplicity, the variance of the slope was therefore neglected in these regressions.

**If the seasonality shown in Figures 5 and 7 is real (signal truly larger than uncertainty) that is an interesting finding.**

Glad you found something interesting about the manuscript. Concerning the "if": Please consider Fig. 7a where you find the S in UT of tropospheric origin together with the UT concentrations during "background" and "moderate influence from tropical volcanism". The latter two sets of data are straightforward averages of measurements in the UT obtained without any regression. First look at the first half of the year. The average of the slightly volcanically influenced case clearly exceeds the background case as a result of the added volcanic aerosol. We know that the Junge layer exists also during background conditions and gives rise to transport of S in this half of the year. In Fig. 6c (or, if you suspect circular evidence, in Fig. 6a in Friberg et al., Tellus 2014) you can see that the tropical volcanism in 2005 - mid 2008 almost doubled the S column compared the background conditions (1999 - 2002), and in Fig. 7b you can see that approximately a factor of 2 appears between the two cases in the first half of the year. Consequently, the tropospheric component should be clearly below the background case during this time of the year, otherwise our results would disagree. In the fall, we see that the volcanic and background data almost coincide (Fig. 7a). Although the stratospheric concentrations are elevated they do not affect the UT concentration very much (Fig. 4b), since we are not in the main season of the deep Brewer-Dobson transport to LMS and UT. It is therefore no surprise that the tropospheric component is more important in the fall. The surprise for us was that we, in addition to the spring maximum, have a second maximum in S concentration of the UT in the fall, a result we obtained by direct averaging of the data taken in the UT. The results of the part of the analysis that you criticize fits very well into the frame obtained by simpler means. We have clearly explained the methods used and errors involved in the different computations. Further explanations have been added to this second revision of the manuscript, see above. There are no reasons to question our results.

Going to CALIOP data to try to explain the fall peak in tropospheric influence on S in UT is also a good idea, but I am not sure how much support the progressive descent of the ATAL in the top row of Fig 8 provides to the athors interpretation of CARIBIC UT/LS S. How much does the ATAL spread zonally over just a few months? Would the upper level anticyclone associated with the Asian monsoon not keep the enhanced aerosol more or less over Tibet?

Have a look in Figs. 8 f - j and Yu et al. in PNAS 2017 (www.pnas.org/cgi/doi/10.1073/pnas.1701170114) for answers to your questions. Here is a quote from the Yu et al. paper: "Between August and December a large portion of ASM aerosol moves poleward following the lower branch of the Brewer–Dobson circulation. The aerosol is eventually flushed out of the stratosphere at high latitudes." (ASM = Asian Summer Monsoon). We take the liberty to interpret "high latitudes" in the meaning high latitude compared to the initial position of the ATAL, because it is well established that the main transport out of the LMS occurs at midlatitudes. With this clarification you see the full agreement between this statement and Fig. 8f-j as well as the explanations given in the manuscript.

Did many (any) of the CARIBIC flight pass through or near the 60-120E longitude band shown in Fig 8?

20% of the samples were taken east of  $60^{\circ}$  E. The entire ATAL region extends to approximately  $15^{\circ}$  E (J. Geophys. Res. Atmos., 120, doi:10.1002/2014JD022372.). 53% of our samples were taken east of  $15^{\circ}$  E. It should, however be pointed out that the effluents of the Nabro eruption occupied this region in 2011, see Fig. 3 in Bourassa et al. (Science 2012). In the manuscript we demonstrate the transport down to flight altitude, both by Caribic and Calipso measurements.

In response to this comment we clarified in line 443: "We therefore illustrate the subsidence using the eruption of the tropical volcano Nabro in the summer 2011 (Table 1).", by adding '... 2011 (Table 1), with effluents that occupied approximately the same region as the ATAL, see Fig. 3 in Bourassa et al. (2012).'

In summary, I am still not convinced that the data analysis is robust enough to believe that from now forward every CARIBIC S measurement can be converted into an estimate of the S column 3 km above the tropopause (Fig 6), or that the source of S in the UT changes from almost completely stratospheric in FM to about 90% tropospheric in ON (Fig. 7). Authors need to use and describe an objective means to bin their data, and more clearly discuss how confident they are about the uncertainty in their derived estimates of "particulate sulfur concentration of tropospheric origin" (also known as the intercepts of the weighted regressions).

As pointed out several times already, we are very confident in the methodology used. We admit that a minor variance component was overlooked, but its effect is insignificant. The error estimates are obtained based on a solid methodology.

Minor editorial comments

Line 82 what is meant by "crust and particle-bound water"?

changed to: "crustal particles and enhanced signal caused by hygroscopic growth"

Lines 147-149 reword this to make it clear what is meant by "not temperature decrease in relation to surface of the earth or the degree of cloud processing" and how either of these is related to distance below the PV tropopause.

changed to: "..., and hence not an altitude-typical degree of cloud processing."

Lines 178-194 This section is where the data binning and how 4 or 5 groups were objectively defined needs to be explained.

Please see the answer on this subject above.

Line 223 and equation 2, seems that year (y) should be group (g). If you really did the regression for 12 seasons and 16 years the number would be >> 52Eqn. 2 is beyond the group stage. The slope depends on the year the sample was taken (mainly

due to volcanism) and the season (air circulation).

Line 284 Fig 4--→Fig 4a

Thank you. Changed in the text.

Line 289 Fig 4-→Fig 4b

Thank you. Changed in the text.

Line 415-417 This sentence is describing the seasonality of "particulate sulfur concentration of tropospheric origin" not the concentration of S in UT. Figure 7 shows nearly no seasonal change in UT S concentration in background year, and a spring peak in the volcanic year.

Thank you. We have changed to: "... distinctive characteristics of the particulate sulfur concentration of tropospheric origin in the UT,"

**Referee 4:**

Review of Paper: Particulate sulfur in the upper troposphere and lowermost stratosphere – sources and climate forcing by Martinsson et al. 2017, submitted to Atmospheric Chemistry and Physics

Martinsson et al. studied sources regions of upper tropospheric sulfur using long-term IAGOS-CARIBIC data. The authors show that the relative contributions (either from transport below or mixed down from the stratosphere) in UT sulfur are dependent on seasons, volcanic activities. The paper has gone through the first round of review with substantial changes based on previous reviewers especially reviewer #1. In general the paper is in good shape, but some concerns (especially on tropospheric contributions) need to be addressed before publication.

a. In the abstract, please give more definition on UT and LMS, i.e. LMS is ~3 km above dynamic tropopause; while UT is how many km below?

The UT in this study is defined by the measurement range of Caribic (99% of the aerosol samples were taken in the altitude range 8.8 - 12 km) and the position of the tropopause. Both the flight and tropopause altitudes vary for the different samples. The part of the LMS regularly probed by

the measurements also defined by the measurement altitude range and the variable tropopause altitude. We add the altitude range of the measurements in the abstract to clarify.

In line 11 we have added: "... stratosphere (LMS) of the northern hemisphere extratropics during monthly intercontinental flights at 8.8 – 12 km altitude of the IAGOS-CARIBIC platform in the time period 1999 – 2014."

We also changed in line 94 from "9 - 12 km" to '8.8 - 12 km' in order to provide the same resolution in data everywhere in the manuscript.

b. Line 25 "We find that tropospheric sources dominate during summer andthe fall as a result of downward transport of the Asian tropopause aerosol layer (ATAL) formed in the Asian monsoon"

The submitted manuscript (Lines 23 - 24) reads: "We find that tropospheric sources dominate during the fall as a result of downward transport of the Asian tropopause aerosol layer (ATAL) formed in the Asian monsoon,". Hence no disagreement with the somewhat more detailed description in line 576.

Line 576 "The ATAL is transported downwards, and affects the extratropical tropopause region in August to December."

These arguments are unclear to me, what does "downwards transport" mean? A recent paper "efficient transport of tropospheric aerosol into the stratosphere via the asian summer monsoon anticyclone" shows that ATAL is mainly lifted above locally to the stratosphere and further transport to high latitudes. See their Figure 4.

First of all (same as in an answer to referee 2): Here is a quote from the Yu et al. paper: "Between August and December a large portion of ASM aerosol moves poleward following the lower branch of the Brewer–Dobson circulation. The aerosol is eventually flushed out of the stratosphere at high latitudes." (ASM = Asian Summer Monsoon). We take the liberty to interpret "high latitudes" in the meaning high latitude compared to the initial position of the ATAL, because it is well established that the main transport out of the LMS occurs at midlatitudes. With this clarification you see the full agreement between this statement and Fig. 8f-j as well as the explanations given in the manuscript.

c. The same paper above argues that in summer/fall seasons, convections transport tracer gases (SO2) and aerosol to UT. Do you think it may be more likely a reason for higher tropospheric contribution of UT sulfur rather than downward transport of ATAL?

As can be seen in Figs. 8b – d, this is part of the ATAL (T for tropopause in ATAL). Fig. 8c covers the same period as Fig. 4 (top) of Vernier et al. (J. Geophys. Res. Atmos., 120, doi:10.1002/2014JD022372.). Just as our Fig. 8c, that of Vernier demonstrates that a large fraction of the ATAL is found below the 380 K isentrope. In that fig. they do not present the tropopause, but in our Fig. 8c you can see that the ATAL reaches all the way to below the tropopause. In the poleward transport the lower branch Brewer-Dobson circulation, as well as transport poleward in the LMS, occurs along isentropes bending towards lower altitudes. In Figs. 8f-j this downward transport can be identified together with a cross-isentrope transport amplifying the downward motion of the air.

We clarify in line 441, replacing "... stratospheric circulation is directed downwards in the extratropics, but the ATAL ..." with "... poleward circulation along isentropes bending downwards, further amplified by an extratropical cross-isentrope, downward component, brings the ATAL down to lower altitudes. The ATAL ...".

d. "SO2 measurements by the satellite-based instrument MIPAS indicate a UT seasonal variation in the NH midlatitudes with low concentrations in December to March and the highest concentrations in June to September (Höpfner et al., 2015)."

In my impression MIPAS SO2 product is of large uncertainties, and may not be meaningful in UT. Shown in Rollins et al. (2017, GRL), MIPAS overestimates the SO2 concentration in TP by a factor of  $\sim$ 3 or so.

We agree that the MIPAS data have large uncertainties (as pointed out also by Höpfner et al. 2015). The large discrepancies between their and Rollins et al. (2017) data are important in a general sense. However, in this manuscript we do not make any use of the numbers, only the seasonal variation reported in the Höpfner et al paper published in Atmos. Chem. Phys. two years ago.

In response we have made the following changes (line 422) from "SO2 measurements by the satellitebased instrument MIPAS indicate a UT seasonal variation in the NH midlatitudes with low concentrations in December to March and the highest concentrations in June to September (Höpfner et al., 2015)."

to 'SO2 measurements by the satellite-based instrument MIPAS have recently become available (Höpfner et al., 2015). These data have large uncertainties (Höpfner et al., 2015), and likely overestimate the SO2 concentration (Rollins et al., 2017). However, here we only qualitatively use the seasonal variation. The MIPAS results indicate a UT seasonal variation in the NH midlatitudes with low concentrations in December to March and the highest concentrations in June to September (Höpfner et al., 2015).'

e. I found the discussions on SO2, CO, OH are difficult to understand. Seems authors claim that direct transport of SO2 and sulfate aerosol from lower troposphere is not going to explain the seasonal variation on tropospheric contribution to UT sulfur, because they will be washed out or cloud-processed. And the authors suggest the concentrations of OH and SO2 are responsible. Well, are we certain that new TP, SO2 oxidation is OH-limited? Any know seasonal cycle on SO2 emissions can explain the seasonal variation? I think some surface SO2 can survive in UT especially in deep convection.

In the manuscript (lines 407 - 414) we discuss the seasonal transport paths from the surface to the UT. Lines 415 - 429 discuss chemical species. The seasonal variation of the common pollution-tracer CO is found to differ from that of particulate sulfur, and we provide one explanation in the strong seasonal cycle of the OH family which produces particulate sulfur and consumes CO. We also discuss the seasonal cycle of the particulate sulfur precursor gas SO2 (from the referenced MIPAS results) and find a covariation between increases of the SO2 and particulate sulfur of tropospheric origin. In this part of the manuscript we try to find the plausible scenario behind our observations, which includes the interpretation that deep convection transports SO2 to the UT

(Lines 424 - 425). Lines 440 - 468 cover the effects of the ATAL, which is also a result of deep convection.

In response we have added (lines 427 - 428): "The increase in CS,UTtrop during the spring and summer months coincides with increases in the convective activity as well as in the concentration of both SO2 and OH, thus offering a plausible explanation." However, the quantitative understanding of the SO2 transport paths and UT seasonality requires further studies.

Co-Editor Decision: Reconsider after minor revisions (Editor review) (02 Aug 2017) by Kostas Tsigaridis Comments to the Author: Dear Authors,

I have another set of review reports available, including a new reviewer, to get one more opinion, since the reports so far did not help me converge to a decision. I believe that the main problem the manuscript has is that the methodology is still fairly opaque, as described in the new reports. Please take into accounts the new comments and submit a revised manuscript.

Please also take into account that one of the reviewers (in a private communication) mentioned that there are still unexamined assumptions about the stratospheric contribution relation with the slopes and intercepts, and a rigorous look at those is needed. The uncertainty in the tropopause determinations is also rather uncertain and the impact of this uncertainty needs to be discussed.

Kind regards, Kostas Tsigaridis

Dear Editor,

We have thoroughly answered the comments from the referees and tried to clarify and improve the legibility of the manuscript on all places we found ways to further improve the manuscript. We hope that you now find the manuscript suitable for publication.

Sincerely, The Authors

**Particulate sulfur in the upper troposphere and lowermost stratosphere - sources and climate forcing**

Bengt G. Martinsson1, Johan Friberg1, Oscar S. Sandvik1, Markus Hermann2, Peter F.J. van Velthoven3 and Andreas Zahn4

1Division of Nuclear Physics, Lund University, Sweden
 2Leibniz Institute for Tropospheric Research, Leipzig, Germany

[revised manuscript text omitted]

bCarn and Prata, 2010 cThomas et al., 2011 dBrühl et al., 2015 cHaywood et al., 2010

750 fClarisse et al., 2012

755